# Fixed-Support Wasserstein Barycenters: Computational Hardness and Fast Algorithm

**Tianyi Lin**[◇]    **Nhat Ho**[†]    **Xi Chen**[‡]    **Marco Cuturi**[◁,▷]    **Michael I. Jordan**[◇]

University of California, Berkeley[◇]

University of Texas, Austin[†]

Stern School of Business, New York University[‡]

CREST - ENSAE[◁], Google Brain[▷]

{darren_lin,jordan}@cs.berkeley.edu, minhnhat@utexas.edu, xchen3@stern.nyu.edu

cuturi@google.com

## Abstract

We study the fixed-support Wasserstein barycenter problem (FS-WBP), which consists in computing the Wasserstein barycenter of $m$ discrete probability measures supported on a finite metric space of size $n$. We show first that the constraint matrix arising from the standard linear programming (LP) representation of the FS-WBP is *not totally unimodular* when $m \geq 3$ and $n \geq 3$. This result resolves an open question pertaining to the relationship between the FS-WBP and the minimum-cost flow (MCF) problem since it proves that the FS-WBP in the standard LP form is not an MCF problem when $m \geq 3$ and $n \geq 3$. We also develop a provably fast *deterministic* variant of the celebrated iterative Bregman projection (IBP) algorithm, named FASTIBP, with a complexity bound of $\widetilde{O}(mn^{7/3}\varepsilon^{-4/3})$, where $\varepsilon \in (0,1)$ is the desired tolerance. This complexity bound is better than the best known complexity bound of $\widetilde{O}(mn^2\varepsilon^{-2})$ for the IBP algorithm in terms of $\varepsilon$, and that of $\widetilde{O}(mn^{5/2}\varepsilon^{-1})$ from accelerated alternating minimization algorithm or accelerated primal-dual adaptive gradient algorithm in terms of $n$. Finally, we conduct extensive experiments with both synthetic data and real images and demonstrate the favorable performance of the FASTIBP algorithm in practice.

## 1 Introduction

Over the past decade, the Wasserstein barycenter problem [1] (WBP) has served as a foundation for theoretical analysis in a wide range of fields, including economics [12, 14] and physics [11, 16, 53] to statistics [39, 30, 51], image and shape analysis [47, 7, 8] and machine learning [19]. The WBP problem is related to the optimal transport (OT) problem, in that both are based on the Wasserstein distance, but the WBP is significantly harder. It requires the minimization of the sum of Wasserstein distances, and typically considers $m > 2$ probability measures. Its closest relative is the multimarginal optimal transport problem [25], which also compares $m$ measures; see Villani [55] for a comprehensive treatment of OT theory and Peyré and Cuturi [44] for an introduction of OT applications and algorithms.

An ongoing focus of work in both the WBP and the OT problem is the design of fast algorithms for computing the relevant distances and optima and the delineation of lower bounds that capture the computational hardness of these problems [44]. For the OT problem, Cuturi [18] introduced the Sinkhorn algorithm which has triggered significant progress [20, 27, 2, 23, 6, 37, 33, 46, 31, 38]. Variants of the Sinkhorn and Greenkhorn algorithms [2, 23, 37] continue to serve as the baseline approaches in practice. As for the theoretical complexity, the best bound is $\widetilde{O}(n^2\varepsilon^{-1})$ [6, 46, 33, 31]. Moreover, Lin et al. [36] provided a complexity bound for the multimarginal OT problem.

There has been significant effort devoted to the development of fast algorithms in the case of $m > 2$ discrete probability measures [47, 19, 13, 7, 4, 3, 52, 57, 10, 45, 15, 54, 22, 56, 34, 32, 29, 26, 9]. This work has provided the foundation for progress on the WBP. An important step forward was the proposal of Cuturi and Doucet [19] to smooth the WBP using an entropic regularization, leading to a simple gradient-descent scheme that was later improved and generalized under the name of the iterative Bregman projection (IBP) algorithm [4, 32]. Further progress includes the semi-dual gradient descent [20, 21], accelerated primal-dual gradient descent (APDAGD) [22, 32], accelerated IBP [29], stochastic gradient descent [15], distributed and parallel gradient descent [52, 54], alternating direction method of multipliers (ADMM) [57, 56] and interior-point algorithm [26]. Very recently, Kroshnin et al. [32] and Guminov et al. [29] have proposed a novel primal-dual framework that made it possible to derive complexity bounds for various algorithms, including IBP, accelerated IBP and APDAGD.

Concerning the computational hardness of the WBP with free support, Anderes et al. [3] proved that the barycenter of $m$ empirical measures is also an empirical measure with support whose cardinality is at most the size of the union of the support of the $m$ measures, minus $m - 1$. When $m = 2$ and the measures are bound and the support is fixed, the computation of the barycenter amounts to solving a network flow problem on a directed graph. Borgwardt and Patterson [9] proved that finding a barycenter of sparse support is NP hard even in the simple setting when $m = 3$. However, their analysis cannot be extended to the fixed-support WBP, where the supports of the constituent $m$ probability measures are prespecified.

**Contribution.** In this paper, we revisit the fixed-support Wasserstein barycenter problem (FS-WBP) between $m$ discrete probability measures supported on a prespecified set of $n$ points. Our contributions can be summarized as follows:

1. We prove that the FS-WBP in the standard LP form is not a minimum-cost flow (MCF) problem in general. In particular, we show that the constraint matrix arising from the standard LP representation of the FS-WBP is totally unimodular when $m \geq 3$ and $n = 2$ but not totally unimodular when $m \geq 3$ and $n \geq 3$. Our results shed light on *the necessity of problem reformulation*—e.g., entropic regularization [19, 4] and block reduction [26].

2. We propose a fast *deterministic* variant of the iterative Bregman projection (IBP) algorithm, named FASTIBP, and provide a theoretical guarantee for the algorithm. Letting $\varepsilon \in (0, 1)$ denote the target tolerance, the complexity bound of the algorithm is $\widetilde{O}(mn^{7/3}\varepsilon^{-4/3})$, which improves the complexity bound of $\widetilde{O}(mn^2\varepsilon^{-2})$ of the IBP algorithm [4] in terms of $\varepsilon$ and the complexity bound of $\widetilde{O}(mn^{5/2}\varepsilon^{-1})$ from the accelerated IBP and APDAGD algorithms in terms of $n$ [32, 29]. We conduct experiments on synthetic and real datasets and demonstrate that the FASTIBP algorithm achieves the favorable performance in practice.

**Organization.** In Section 2, we present the entropic-regularized FS-WBP and the dual problem. In Section 3, we provide our computational hardness results for the FS-WBP in the standard LP form. In Section 4, we propose and analyze the FASTIBP algorithm. We conduct experiments on synthetic and real data in Section 5 and conclude in Section 6. All the proofs are deferred to Appendix.

**Notation.** We let $[n]$ be the set $\{1, \ldots, n\}$ and $\mathbb{R}^n_+$ be all vectors in $\mathbb{R}^n$ with nonnegative components. $\mathbf{1}_n$ and $\mathbf{0}_n$ are the $n$-vectors of ones and zeros. $\Delta^n$ stands for the probability simplex. For a function $f$, we denote $\nabla f$ and $\nabla_\lambda f$ as the full gradient and the gradient with respect to $\lambda$. For $x \in \mathbb{R}^n$ and $1 \leq p \leq \infty$, we write $\|x\|_p$ for its $\ell_p$-norm. For $X \in \mathbb{R}^{n \times n}$, $\text{vec}(X) \in \mathbb{R}^{n^2}$ and $\det(X)$ stand for the vector representation and the determinant. $r(X) = X\mathbf{1}_n$ and $c(X) = X^\top \mathbf{1}_n$. Let $X, Y \in \mathbb{R}^{n \times n}$, the Frobenius and Kronecker inner product are denoted by $\langle X, Y \rangle$ and $X \otimes Y$. Given $n$ and $\varepsilon$, the notation $a = O(b(n, \varepsilon))$ stands for $a \leq C \cdot b(n, \varepsilon)$ where $C > 0$ is independent of $n$ and $\varepsilon$, and $a = \widetilde{O}(b(n, \varepsilon))$ indicates previous inequality where $C$ depends on the logarithmic factors of $n$ and $\varepsilon$.

## 2 Preliminaries and Technical Background

In this section, we introduce the setup of the fixed-support Wasserstein barycenter problem (FS-WBP), starting with the linear programming (LP) presentation and entropic-regularized formulation and including a specification of an approximate barycenter. All the proofs are deferred to Appendix D.

## 2.1 Linear programming formulation

For $p \geq 1$, let $\mathcal{P}_p(\Omega)$ be the set of Borel probability measures on $\Omega$ with finite $p$-th moment. The Wasserstein distance of order $p \geq 1$ between $\mu, \nu \in \mathcal{P}_p(\Omega)$ is defined by [55]:

$$W_p(\mu, \nu) := \inf_{\pi \in \Pi(\mu,\nu)} \left( \int_{\Omega \times \Omega} d^p(\mathbf{x}, \mathbf{y}) \, \pi(d\mathbf{x}, d\mathbf{y}) \right)^{1/p}, \tag{1}$$

where $d(\cdot, \cdot)$ is a metric on $\Omega$ and $\Pi(\mu, \nu)$ is the set of couplings between $\mu$ and $\nu$. Given a weight vector $(\omega_1, \omega_2, \dots, \omega_m) \in \Delta^m$ for $m \geq 2$, the *Wasserstein barycenter* [1] of $m$ probability measures $\{\mu_k\}_{k=1}^m$ is a solution of the following functional minimization problem

$$\min_{\mu \in \mathcal{P}_p(\Omega)} \sum_{k=1}^m \omega_k W_p^p(\mu, \mu_k). \tag{2}$$

Because our goal is to provide computational schemes to approximately solve the WBP, we need to provide a definition of an $\varepsilon$-approximate solution to the WBP.

**Definition 2.1.** *The probability measure $\widehat{\mu} \in \mathcal{P}_p(\Omega)$ is called an $\varepsilon$-approximate barycenter if $\sum_{k=1}^m \omega_k W_p^p(\widehat{\mu}, \mu_k) \leq \sum_{k=1}^m \omega_k W_p^p(\mu^\star, \mu_k) + \varepsilon$ where $\mu^\star$ is an optimal solution to problem* (2).

There are two main settings: (i) *free-support Wasserstein barycenter*, namely, when we optimize both the weights and supports of the barycenter in Eq. (2); and (ii) *fixed-support Wasserstein barycenter*, namely, when the supports of the barycenter are obtained from those from the probability measures $\{\mu_k\}_{k=1}^m$ and we optimize the weights of the barycenter in Eq. (2).

*The free-support WBP problem is notoriously difficult to solve.* It can either be solved using a solution to the multimarginal-OT (MOT) problem, as described in detail by Agueh and Carlier [1], or approximated using alternative optimization techniques. Assuming that each measure is supported on $n$ distinct points, the WBP problem can be solved *exactly* by solving first a MOT, to then compute $(n-1)m + 1$ barycenters of *points* in $\Omega$ (these barycenters are exactly the support of the barycentric measure). Solving a MOT is, however, equivalent to solving an LP with $n^m$ variables and $(n-1)m + 1$ constraints. The other route, alternative optimization, requires specifying an initial guess for the barycenter, a discrete measure supported on $k$ weighted points (where $k$ is predefined). One can then proceed by updating the locations of $\mu$ (or even add new ones) to decrease the objective function in Eq. (2), before changing their weights. In the Euclidean setting with $p = 2$, the free-support WBP is closely related to the clustering problem, and equivalent to $k$-means when $m = 1$ [19]. Whereas solving the free-support WBP using MOT results in a convex (yet intractable) problem, the alternating mimimization approach is not, in very much the same way that the $k$-means problem is not, and results in the minimization of a piece-wise quadratic function. *On the other hand, the fixed-support WBP is comparatively easier to solve, and as such has played a role in real-world applications.* For instance, in imaging sciences, pixels and voxels are supported on a predefined, finite grid. In these applications, the barycenter and $\mu_k$ measures share the same support.

In view of this, throughout the remainder of the paper, we let $(\mu_k)_{k=1}^m$ be discrete probability measures and take the support points $\{\mathbf{x}_i^k\}_{i \in [n]}$ to be fixed. Since $\{\mu_k\}_{k=1}^m$ have the fixed support, they are fully characterized by the weights $\{u^k\}_{k=1}^m$. Accordingly, the support of the barycenter $\{\widehat{\mathbf{x}}_i\}_{i \in [n]}$ is also fixed and can be prespecified by $\{\mathbf{x}_i^k\}_{i \in [n]}$. Given this setup, the FS-WBP between $\{\mu_k\}_{k=1}^m$ has the following standard LP representation [19, 4, 44]:

$$\min_{\{X_i\}_{i=1}^m \subseteq \mathbb{R}_+^{n \times n}} \sum_{k=1}^m \omega_k \langle C_k, X_k \rangle, \quad \text{s.t.} \quad \begin{array}{l} r(X_k) = u^k \text{ for all } k \in [m], \\ c(X_{k+1}) = c(X_k) \text{ for all } k \in [m-1]. \end{array} \tag{3}$$

where $\{X_k\}_{k=1}^m$ and $\{C_k\}_{k=1}^m \subseteq \mathbb{R}_+^{n \times \dots \times n}$ denote a set of *transportation plans* and *nonnegative cost matrices* and $(C_k)_{ij} = d^p(\mathbf{x}_i^k, \widehat{\mathbf{x}}_j)$ for all $k \in [m]$. The fixed-support Wasserstein barycenter $u \in \Delta_n$ is determined by the weight $\sum_{k=1}^m \omega_k c(X_k)$ and the support $(\widehat{\mathbf{x}}_1, \widehat{\mathbf{x}}_2, \dots, \widehat{\mathbf{x}}_n)$.

From Eq. (3), the FS-WBP is an LP with $2mn - n$ equality constraints and $mn^2$ variables. This has inspired work on solving the FS-WBP using classical optimization algorithms [26, 56]. Although progress has been made, the understanding of the structure of FS-WBP via this approach has remained limited. Particularly, while the OT problem [55] is equivalent to a minimum-cost flow (MCF) problem, it remains unknown whether the FS-WBP is a MCF problem even in the simplest setting when $m = 2$.

## 2.2 Entropic regularized FS-WBP

Using Cuturi's entropic approach to the OT problem [18], we define a regularized version of the FS-WBP in Eq. (3), where an entropic regularization term is added to the Wasserstein barycenter objective. The resulting formulation is as follows:

$$\min_{\{X_i\}_{i=1}^m \subseteq \mathbb{R}_+^{n \times n}} \sum_{k=1}^m \omega_k(\langle C_k, X_k \rangle - \eta H(X_k)), \quad \text{s.t.} \quad \begin{array}{l} r(X_k) = u^k \text{ for all } k \in [m], \\ c(X_{k+1}) = c(X_k) \text{ for all } k \in [m-1]. \end{array} \quad (4)$$

where $\eta > 0$ is the parameter and $H(X) := -\langle X, \log(X) - \mathbf{1}_n \mathbf{1}_n^\top \rangle$ denotes the entropic regularization term. We refer to Eq. (4) as *entropic regularized FS-WBP*. When $\eta$ is large, the optimal value of entropic regularized FS-WBP may yield a poor approximation of the cost of the FS-WBP. To guarantee a good approximation, we scale the parameter $\eta$ as a function of the desired accuracy.

**Definition 2.2.** *The probability vector $\widehat{u} \in \Delta^n$ is called an $\varepsilon$-approximate barycenter if there exists a feasible solution $(\widehat{X}_1, \widehat{X}_2, \ldots, \widehat{X}_m) \in \mathbb{R}_+^{n \times n} \times \cdots \times \mathbb{R}_+^{n \times n}$ for the FS-WBP in Eq. (3) such that $\widehat{u} = \sum_{k=1}^m \omega_k c(\widehat{X}_k)$ for all $k \in [m]$ and $\sum_{k=1}^m \omega_k \langle C_k, \widehat{X}_k \rangle \leq \sum_{k=1}^m \omega_k \langle C_k, X_k^\star \rangle + \varepsilon$ where $(X_1^\star, X_2^\star, \ldots, X_m^\star)$ is an optimal solution of the FS-WBP in Eq. (3).*

With these definitions in mind, we develop efficient algorithms for approximately solving the FS-WBP where the dependence on $m$, $n$ and $\varepsilon$ is competitive to state-of-the-art algorithms [32, 29].

## 2.3 Dual entropic regularized FS-WBP

Using the duality theory of convex optimization [48], one dual form of entropic regularized FS-WBP has been derived before [19, 32]. Differing from the usual 2-marginal or multimarginal OT [21, 36], the dual entropic regularized FS-WBP is a convex optimization problem with an affine constraint set. Formally, we have

$$\min_{\lambda, \tau \in \mathbb{R}^{mn}} \varphi_{\text{old}}(\lambda, \tau) := \sum_{k=1}^m \omega_k \left( \sum_{1 \leq i,j \leq n} e^{\lambda_{ki} + \tau_{kj} - \eta^{-1}(C_k)_{ij}} - \lambda_k^\top u^k \right), \quad \text{s.t.} \sum_{k=1}^m \omega_k \tau_k = \mathbf{0}_n. \quad (5)$$

However, the objective function in Eq. (5) is not sufficiently smooth because of the sum of exponents. This makes the acceleration very challenging. In order to alleviate this issue, we turn to another smooth dual form of entropic-regularized FS-WBP as follows,

$$\min_{\lambda, \tau \in \mathbb{R}^{mn}} \varphi(\lambda, \tau) := \sum_{k=1}^m \omega_k \left( \log(\|B_k(\lambda_k, \tau_k)\|_1) - \lambda_k^\top u^k \right), \quad \text{s.t.} \sum_{k=1}^m \omega_k \tau_k = \mathbf{0}_n. \quad (6)$$

We call it the *dual entropic-regularized FS-WBP problem* and refer the interested reader to Appendix A for a complete derivation of Eq. (6).

**Remark 2.1.** *The first part of the objective function $\varphi$ is in the form of the logarithm of sum of exponents while the second part is a linear function. This is different from the objective function used in previous dual entropic regularized OT problem in Eq. (5). We also note that Eq. (6) is a special instance of a softmax minimization problem, and the objective function $\varphi$ is known to be smooth [40]. Finally, we point out that the same problem was derived in the concurrent work by Guminov et al. [29] and used for analyzing the accelerated alternating minimization algorithm.*

## 2.4 Properties of dual entropic regularized FS-WBP

In this section, we present several useful properties of the dual entropic regularized MOT in Eq. (6). In particular, there exists an optimal solution $(\lambda^\star, \tau^\star)$ which has an upper bound in $\ell_\infty$-norm.

**Lemma 2.2.** *For the dual entropic regularized FS-WBP, let $\bar{C} = \max_{1 \leq k \leq m} \|C_k\|_\infty$ and $\bar{u} = \min_{1 \leq k \leq m, 1 \leq j \leq n} u_{kj}$, there exists an optimal solution $(\lambda^\star, \tau^\star)$ such that the following $\ell_\infty$-norm bound holds true:*

$$\|\lambda_k^\star\|_\infty \leq R_\lambda, \quad \|\tau_k^\star\|_\infty \leq R_\tau, \quad \text{for all } k \in [m], \quad (7)$$

*where $R_\lambda = 5\eta^{-1}\bar{C} + \log(n) - \log(\bar{u})$ and $R_\tau = 4\eta^{-1}\bar{C}$.*

**Remark 2.3.** *Lemma 2.2 is analogous to [37, Lemma 3.2] for the OT problem. However, the dual entropic-regularized FS-WBP is more complex and requires a novel constructive iterate, $(\lambda^\star, \tau^\star) = \sum_{j=1}^m \omega_j(\lambda^j, \tau^j)$. Moreover, the techniques in Kroshnin et al. [32] are not applicable for the analysis of the FASTIBP algorithm, and, accordingly, Lemma 2.2 is crucial for the analysis.*

The upper bound for the $\ell_\infty$-norm of the optimal solution of dual entropic-regularized FS-WBP in Lemma 2.2 leads to the following straightforward consequence.

**Corollary 2.4.** *For the dual entropic regularized FS-WBP, there exists an optimal solution $(\lambda^\star, \tau^\star)$ such that for all $k \in [m]$,*

$$\|\lambda_k^\star\| \leq \sqrt{n} R_\lambda, \quad \|\tau_k^\star\| \leq \sqrt{n} R_\tau, \quad \text{for all } k \in [m], \tag{8}$$

*where $R_\lambda, R_\tau > 0$ are defined in Lemma 2.2.*

Finally, $\varphi$ can be decomposed into the weighted sum of $m$ functions and prove that each component function $\varphi_k$ has Lipschitz continuous gradient with the constant 4 in the following lemma.

**Lemma 2.5.** *The following statement holds true, $\varphi(\lambda, \tau) = \sum_{k=1}^m \varphi_k(\lambda_k, \tau_k)$ where $\varphi_k(\lambda_k, \tau_k) = \log(\boldsymbol{I}_n^\top B_k(\lambda_k, \tau_k)\boldsymbol{I}_n) - \lambda_k^\top u^k$ for all $k \in [m]$. Moreover, each $\varphi_k$ has Lipschitz continuous gradient in $\ell_2$-norm and the Lipschitz constant is upper bounded by 4. Formally,*

$$\varphi(\lambda', \tau') - \varphi(\lambda, \tau) \leq \begin{pmatrix} \lambda' - \lambda \\ \tau' - \tau \end{pmatrix}^\top \nabla\varphi(\lambda, \tau) + 2 \left( \sum_{k=1}^m \omega_k \left\| \begin{pmatrix} \lambda_k' - \lambda_k \\ \tau_k' - \tau_k \end{pmatrix} \right\|^2 \right) \quad \text{for all } k \in [m]. \tag{9}$$

**Remark 2.6.** *It is worthy noting that Lemma 2.5 exploits the decomposable structure of the dual function $\varphi$, and gives the a **weighted** smoothness inequality. This inequality is necessary for deriving the complexity bound which depends linearly on the number of probability measures.*

# 3 Computational Hardness

In this section, we analyze the computational hardness of the FS-WBP in Eq. (3). After introducing some characterization theorems in combinatorial optimization, we show that the FS-WBP in Eq. (3) is a minimum-cost flow (MCF) problem when $m = 2$ and $n \geq 3$ but not when $m \geq 3$ and $n \geq 3$. We refer the interested readers to Appendix B for the definition of MCF problem, Appendix C for an illustrative example which explicitly analyzes the constraint matrix of Eq. (3) when $m = n = 3$, and Appendix D for the missing proofs.

## 3.1 Combinatorial techniques

We present some classical results in combinatorial optimization and graph theory, including Ghouila-Houri's celebrated characterization theorem [28].

**Definition 3.1.** *A totally unimodular (TU) matrix is one for which every square submatrix has determinant $-1$, 0 or 1.*

**Proposition 3.1** (Ghouila-Houri). *A $\{-1, 0, 1\}$-valued matrix $A \in \mathbb{R}^{m \times n}$ is TU if and only if for each $I \subseteq [m]$ there is a partition $I = I_1 \cup I_2$ so that $\sum_{i \in I_1} a_{ij} - \sum_{i \in I_2} a_{ij} \in \{-1, 0, 1\}$ for $j \in [n]$.*

**Proposition 3.2.** *[5, Theorem 1, Chapter 15] Let $A$ be a $\{-1, 0, 1\}$-valued matrix. Then $A$ is TU if each column contains at most two nonzero entries and all rows are partitioned into two sets $I_1$ and $I_2$ such that: If two nonzero entries of a column have the same sign, they are in different sets. If these two entries have different signs, they are in the same set.*

**Proposition 3.3.** *The constraint matrix arising in a MCF problem is TU and its rows are categorized into a single set using Proposition 3.2.*

## 3.2 Main result

We present our computational hardness of the FS-WBP in Eq. (3). First, the FS-WBP in this LP form is an MCF problem when $m = 2$ and $n \geq 2$. Indeed, it is a transportation problem with $n$ warehouses, $n$ transshipment centers and $n$ retail outlets. Each $u_{1i}$ is the amount of supply provided by the $i$th warehouse and each $u_{2j}$ is the amount of demand requested by the $j$th retail outlet. $(X_1)_{ij}$ is the flow sent from $i$th warehouse to the $j$th transshipment center and $(X_2)_{ij}$ is the flow sent from the $i$th transshipment center to the $j$th retail outlet. $(C_1)_{ij}$ and $(C_2)_{ij}$ refer to the unit cost of the corresponding flow. See [3, Page 400].

Proceed to the setting $m \geq 3$, it is unclear whether the graph representation of the FS-WBP carries over. Instead, we turn to algebraic techniques and provide an explicit form as follows:

$$\min \sum_{k=1}^{m} \langle C_k, X_k \rangle \quad \text{s.t.} \quad \begin{pmatrix} -E & \cdots & \cdots & \cdots & & \cdots \\ \vdots & E & \ddots & & \ddots & \vdots \\ \vdots & \ddots & \ddots & & \ddots & \vdots \\ \vdots & \ddots & \ddots & (-1)^{m-1}E & & \vdots \\ \vdots & \ddots & \ddots & & \ddots & (-1)^m E \\ G & -G & \ddots & & \ddots & \vdots \\ \vdots & -G & G & & \ddots & \vdots \\ \vdots & \ddots & \ddots & & \ddots & \vdots \\ \cdots & \cdots & \cdots & (-1)^m G & & (-1)^{m+1}G \end{pmatrix} \begin{pmatrix} \mathrm{vec}\,(X_1) \\ \mathrm{vec}\,(X_2) \\ \vdots \\ \vdots \\ \vdots \\ \vdots \\ \mathrm{vec}\,(X_m) \end{pmatrix} = \begin{pmatrix} -u^1 \\ u^2 \\ \vdots \\ (-1)^{m-1}u^{m-1} \\ (-1)^m u^m \\ \mathbf{0}_n \\ \vdots \\ \mathbf{0}_n \end{pmatrix},$$

(10)

where $E = I_n \otimes \mathbf{1}_n^\top \in \mathbb{R}^{n \times n^2}$ and $G = \mathbf{1}_n^\top \otimes I_n \in \mathbb{R}^{n \times n^2}$. Each column of the constraint matrix arising in Eq. (10) has either 2 or 3 nonzero entries in $\{-1, 0, 1\}$. In the following theorem, we study the structure of the constraint matrix when $m \geq 3$ and $n = 2$.

**Theorem 3.4.** *The constraint matrix arising in Eq.* (10) *is TU when $m \geq 3$ and $n = 2$.*

**Theorem 3.5.** *The FS-WBP in Eq.* (10) *is not a MCF problem when $m \geq 3$ and $n \geq 3$.*

**Remark 3.6.** *Theorem 3.5 resolves an open question and partially explains why the direct application of network flow algorithms to the FS-WBP in Eq.* (10) *is inefficient. However, this does not eliminate the possibility that the FS-WBP is equivalent to some other LP with good complexity. For example, Ge et al. [26] have recently successfully identified an equivalent LP formulation of the FS-WBP which is suitable for the interior-point algorithm. Furthermore, our results support the problem reformulation of the FS-WBP which forms the basis for various algorithms; e.g., Benamou et al. [4], Cuturi and Peyré [20], Kroshnin et al. [32], Ge et al. [26], Guminov et al. [29].*

## 4 Fast Iterative Bregman Projection

In this section, we propose a fast *deterministic* variant of the iterative Bregman projection (IBP) algorithm, named FASTIBP, with the complexity bound of $\widetilde{O}(mn^{7/3}\varepsilon^{-4/3})$; see the pseudocode in Algorithm 1 and 2. Note that $(B_1(\lambda_1^t, \tau_1^t), \ldots, B_m(\lambda_m^t, \tau_m^t))$ stand for the primal variables while $(\lambda^t, \tau^t)$ are the dual variables for the entropic regularized FS-WBP. Due to the space limit, we defer all the technical lemmas and proofs to Appendix E.

While the IBP algorithm can be interpreted as a dual coordinate descent, the acceleration achieved by the FASTIBP algorithm mostly depends on the refined characterization of per-iteration progress using the scheme with momentum; see **Step 1-3** and **Step 8**. This scheme has been studied by [41, 42, 24, 43] yet *first* introduced to accelerate the optimal transport algorithms. Furthermore, **Step 4** guarantees that $\{\varphi(\check{\lambda}^t, \check{\tau}^t)\}_{t \geq 0}$ is monotonically decreasing and **Step 7** ensures the sufficient large progress from $(\lambda_k^t, \tau_k^t)$ to $(\check{\lambda}^{t+1}, \check{\tau}^{t+1})$. **Step 5** are performed such that $\tau_k^t = \check{\tau}_k^t$ satisfies the bounded difference property: $\max_{1 \leq i \leq n}(\tau_k^t)_i - \min_{1 \leq i \leq n}(\tau_k^t)_i \leq R_\tau/2$ while **Step 6** guarantees that the marginal conditions hold true: $r(B_k(\lambda_k^t, \tau_k^t)) = u^k$ for all $k \in [m]$. We see from Guminov et al. [29, Lemma 9] that **Step 5-7** refer to the alternating minimization steps for the dual objective function $\varphi$ with respect to two-block variable $(\lambda, \tau)$. We remark that **Step 4-7** are *specialized* to the FS-WBP in Eq. (3) and have *not* appeared in the coordinate descent literature.

Finally, the optimality conditions of primal entropic regularized WBP in Eq. (4) and dual entropic regularized WBP in Eq. (6) are

$$\frac{r(B_k(\lambda_k, \tau_k))}{\|B_k(\lambda_k, \tau_k)\|_1} - u^k = \frac{c(B_k(\lambda_k, \tau_k))}{\|B_k(\lambda_k, \tau_k)\|_1} - \sum_{i=1}^{m} \omega_i \frac{c(B_i(\lambda_i, \tau_i))}{\|B_i(\lambda_i, \tau_i)\|_1} = \mathbf{0}_n \text{ for all } k \in [m], \quad \sum_{k=1}^{m} \omega_k \tau_k = \mathbf{0}_n.$$

---

**Algorithm 1:** FASTIBP($\{C_k, u^k\}_{k \in [m]}, \varepsilon$)

---

**Initialization:** $t = 0, \theta_0 = 1$ and $\check{\lambda}^0 = \tilde{\lambda}^0 = \check{\tau}^0 = \tilde{\tau}^0 = \mathbf{0}_{mn}$.

**while** $E_t > \varepsilon$ **do**

  **Step 1:** Compute $\begin{pmatrix} \bar{\lambda}^t \\ \bar{\tau}^t \end{pmatrix} = (1 - \theta_t) \begin{pmatrix} \check{\lambda}^t \\ \check{\tau}^t \end{pmatrix} + \theta_t \begin{pmatrix} \tilde{\lambda}^t \\ \tilde{\tau}^t \end{pmatrix}$.

  **Step 2:** Compute $r_k = r(B_k(\bar{\lambda}_k^t, \bar{\tau}_k^t))$ and $c_k = c(B_k(\bar{\lambda}_k^t, \bar{\tau}_k^t))$ for all $k \in [m]$ and perform

$$\tilde{\lambda}_k^{t+1} = \tilde{\lambda}_k^t - \frac{1}{4\theta_k} \left( \frac{r_k}{\mathbf{1}_n^\top r_k} - u^k \right), \quad \text{for all } k \in [m],$$

$$\tilde{\tau}^{t+1} = \underset{\sum_{k=1}^m \omega_k \tau_k = \mathbf{0}_n}{\operatorname{argmin}} \sum_{k=1}^m \omega_k \left[ (\tau_k - \tilde{\tau}_k^t)^\top \frac{c_k}{\mathbf{1}_n^\top c_k} + 2\theta_t \|\tau_k - \tilde{\tau}_k^t\|^2 \right].$$

  **Step 3:** Compute $\begin{pmatrix} \widehat{\lambda}^t \\ \widehat{\tau}^t \end{pmatrix} = \begin{pmatrix} \bar{\lambda}^t \\ \bar{\tau}^t \end{pmatrix} + \theta_t \begin{pmatrix} \tilde{\lambda}^{t+1} \\ \tilde{\tau}^{t+1} \end{pmatrix} - \theta_t \begin{pmatrix} \tilde{\lambda}^t \\ \tilde{\tau}^t \end{pmatrix}$.

  **Step 4:** Compute $\begin{pmatrix} \acute{\lambda}^t \\ \acute{\tau}^t \end{pmatrix} = \operatorname{argmin} \left\{ \varphi(\lambda, \tau) \mid \begin{pmatrix} \lambda \\ \tau \end{pmatrix} \in \left\{ \begin{pmatrix} \check{\lambda}^t \\ \check{\tau}^t \end{pmatrix}, \begin{pmatrix} \widehat{\lambda}^t \\ \widehat{\tau}^t \end{pmatrix} \right\} \right\}$.

  **Step 5a:** Compute $c_k = c(B_k(\acute{\lambda}_k^t, \acute{\tau}_k^t))$ for all $k \in [m]$.
  **Step 5b:** Compute $\dot{\tau}_k^t = \acute{\tau}_k^t + \sum_{k=1}^m \omega_k \log(c_k) - \log(c_k)$ for all $k \in [m]$ and $\grave{\lambda}^{t+1} = \acute{\lambda}^t$.
  **Step 6a:** Compute $r_k = r(B_k(\grave{\lambda}_k^t, \dot{\tau}_k^t))$ for all $k \in [m]$.
  **Step 6b:** Compute $\lambda_k^t = \grave{\lambda}_k^t + \log(u^k) - \log(r_k)$ for all $k \in [m]$ and $\tau^t = \dot{\tau}^t$.
  **Step 7a:** Compute $c_k = c(B_k(\lambda_k^t, \tau_k^t))$ for all $k \in [m]$.
  **Step 7b:** Compute $\check{\tau}_k^{t+1} = \tau_k^t + \sum_{k=1}^m \omega_k \log(c_k) - \log(c_k)$ for all $k \in [m]$ and $\check{\lambda}^{t+1} = \lambda^t$.
  **Step 8:** Compute $\theta_{t+1} = \theta_t(\sqrt{\theta_t^2 + 4} - \theta_t)/2$.
  **Step 9:** Increment by $t = t + 1$.

**end while**

**Output:** $(B_1(\lambda_1^t, \tau_1^t), B_2(\lambda_2^t, \tau_2^t), \ldots, B_m(\lambda_m^t, \tau_m^t))$.

---

**Algorithm 2:** Finding a Wasserstein barycenter by the FASTIBP algorithm

---

**Input:** $\eta = \varepsilon/(4 \log(n))$ and $\bar{\varepsilon} = \varepsilon/(4 \max_{1 \le k \le m} \|C_k\|_\infty)$.

**Step 1:** Compute $(\tilde{u}^1, \ldots, \tilde{u}^m) = (1 - \bar{\varepsilon}/4)(u^1, \ldots, u^m) + (\bar{\varepsilon}/4n)(\mathbf{1}_n, \ldots, \mathbf{1}_n)$.

**Step 2:** Compute $(\widetilde{X}_1, \widetilde{X}_2, \ldots, \widetilde{X}_m) = \text{FASTIBP}(\{C_k, \tilde{u}^k\}_{k \in [m]}, \bar{\varepsilon}/2)$.

**Step 3:** Round $(\widetilde{X}_1, \widetilde{X}_2, \ldots, \widetilde{X}_m)$ to $(\widehat{X}_1, \widehat{X}_2, \ldots, \widehat{X}_m)$ using Kroshnin et al. [32, Algorithm 4] such that $(\widehat{X}_1, \widehat{X}_2, \ldots, \widehat{X}_m)$ is feasible to the FS-WBP in Eq. (3).

**Step 4:** Compute $\widehat{u} = \sum_{k=1}^m \omega_k \widehat{X}_k^\top \mathbf{1}_n$

**Output:** $\widehat{u}$.

---

Since the FASTIBP algorithm guarantees that $\sum_{k=1}^m \omega_k \tau_k^t = \mathbf{0}_n$ and $r(B_k(\lambda_k^t, \tau_k^t)) = u^k \in \Delta^n$ for all $k \in [m]$, the criterion depends on the following quantity to measure the residue at each iteration:

$$E_t := \sum_{k=1}^m \omega_k \| c(B_k(\lambda_k^t, \tau_k^t)) - \sum_{i=1}^m \omega_i c(B_i(\lambda_i^t, \tau_i^t)) \|_1. \tag{11}$$

While the existing algorithms, e.g., accelerated IBP and APDAGD, are developed based on the primal-dual framework which allows for better dependence on $1/\varepsilon$ by directly optimizing $E_t$, the FASTIBP algorithm indirectly optimizes $E_t$ through the dual objective gap and the scheme with momentum (cf. **Step 1-3** and **Step 8**), which can lead to better dependence on $n$.

**Remark 4.1.** *First, we notice that each iteration updates $O(mn^2)$ entries while $\tilde{\lambda}$ and $\check{\lambda}$ can be efficiently updated in distributed manner. Second, $4m$ marginals can be updated effectively by using $r(e^{-\eta^{-1}C_k})$ and $c(e^{-\eta^{-1}C_k})$ for all $k \in [m]$, which are stored before main loop. Finally, $(\widehat{X}_1, \ldots, \widehat{X}_m)$ are approximate optimal transportation plans between $m$ measures $\{u^k\}_{k \in [m]}$ and $\varepsilon$-approximate barycenter $\widehat{u}$. We can also construct $\widehat{u}$ by using $(\widetilde{X}_1, \ldots, \widetilde{X}_m)$; see [32, Section 2.2].*

**Theorem 4.2.** *Let $\{(\lambda^t, \tau^t)\}_{t \ge 0}$ be the iterates generated by the FASTIBP algorithm. The number of iterations required to reach the stopping criterion $E_t \le \varepsilon$ satisfies $t \le 1 + 10(n(R_\lambda^2 + R_\tau^2)\varepsilon^{-2})^{1/3}$ where $R_\lambda, R_\tau > 0$ are defined in Lemma 2.2.*

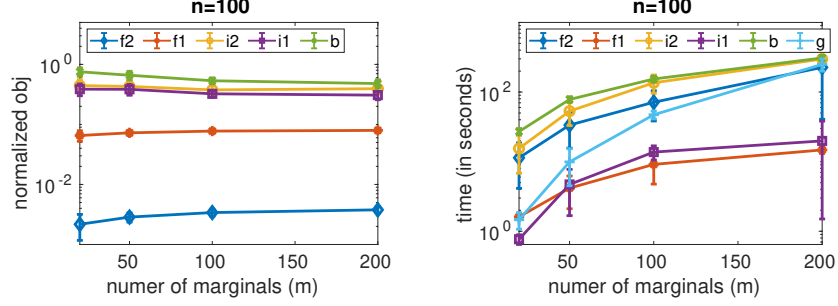

Figure 1: The average normalized objective value and computational time (in seconds) of FASTIBP, IBP, BADMM, and Gurobi from 10 independent trials.

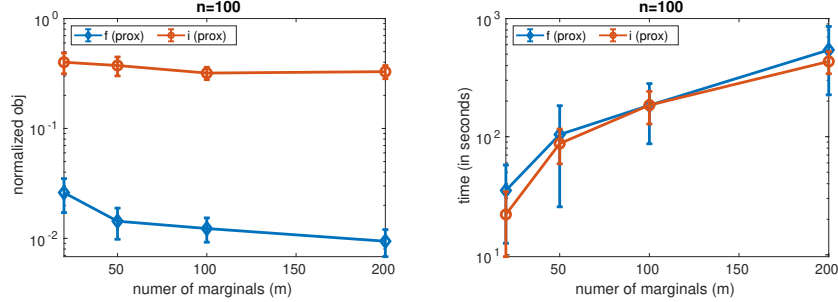

Figure 2: The average normalized objective value and computational time (in seconds) of the proximal variants of FASTIBP and IBP from 10 independent trials.

Equipped with the result of Theorem 4.2, we are ready to present the complexity bound of Algorithm 2 for approximating the FS-WBP in Eq. (3).

**Theorem 4.3.** *The* FASTIBP *algorithm for approximately solving the FS-WBP in Eq. (3) (Algorithm 2) returns an $\varepsilon$-approximate barycenter $\widehat{u} \in \mathbb{R}^n$ within*

$$O\left(mn^{7/3}\left(\frac{(\max_{1 \leq k \leq m} \|C_k\|_\infty)\sqrt{\log(n)}}{\varepsilon}\right)^{4/3}\right)$$

*arithmetic operations.*

**Remark 4.4.** *We assume for simplicity that all measures have the same support size . This assumption is not necessary and our analysis is still valid when each measure has fixed support of different size. However, our results can not be generalized to the free-support Wasserstein barycenter problem in general since the computation of free-support barycenters requires solving a multimarginal OT problem where the complexity bounds of algorithms become much worse; see Lin et al. [36].*

## 5 Experiments

In this section, we evaluate the FASTIBP algorithm for computing fixed-support Wasserstein barycenters. In all our experiments, we consider the Wasserstein distance with $\ell_2$-norm and compare our algorithm with Gurobi, iterative Bregman projection (IBP) algorithm [4] and Bregman ADMM (BADMM) [57][1]. In our figures, "g" stands for Gurobi; "b" stands for BADMM; "i1" and "i2" stand for IBP with $\eta = 0.01$ and $\eta = 0.001$; "f1" and "f2" stand for the FASTIBP algorithm with $\eta = 0.01$ and $\eta = 0.001$. Due to space limits, some details and results are deferred to Appendix F and G.

**Synthetic data.** We present some preliminary numerical results in Figure 1 and 2. Given $n = 100$, we evaluate the performance of FASTIBP, IBP, BADMM algorithms, and Gurobi by varying $m \in \{20, 50, 100, 200\}$ and use the same setup to compare the proximal variants of FASTIBP and IBP. We use the proximal framework [32] with the same parameter setting as provided by their paper.

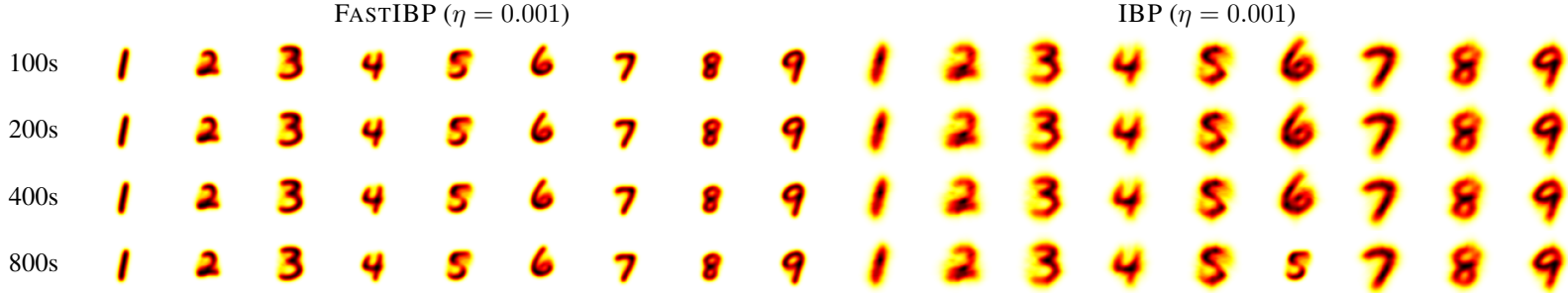

Table 1: Approximate barycenters obtained by running FASTIBP and IBP for 100s, 200s, 400s, 800s.

From Figure 1, the FASTIBP algorithm performs better than BADMM and IBP in the sense that it consistently returns an objective value closer to that of Gurobi in less computational time. IBP converges very fast when $\eta = 0.01$, but suffers from a crude solution with poor objective value; BADMM takes more time with unsatisfactory objective value, and is not provably convergent in theory; Gurobi solves the problem of relatively small size very efficiently but suffer from scalability. From Figure 2, the proximal variant of FASTIBP algorithm outperforms that of IBP algorithm in terms of objective value while not sacrificing the efficiency.

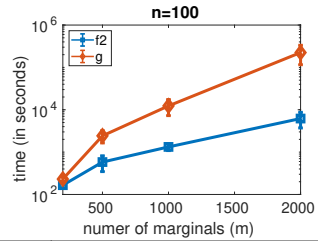

To further compare Gurobi and the FASTIBP algorithm, we conduct the experiment with $n = 100$ and the varying number of marginals $m \in \{200, 500, 1000, 2000\}$. We fix $\mathrm{Tol}_{\mathsf{fibp}} = 10^{-6}$ but without setting the maximum iteration number. Figure 3 shows the average running time taken by two algorithms over 5 independent trials. We see that the FASTIBP algorithm is competitive with Gurobi in terms of objective value and feasibility violation. In terms of computational time, the FASTIBP algorithm increases linearly with respect to the number of marginals, while Gurobi increases much more rapidly. Compared to the similar results of Gurobi presented before [56, 26], we find that the feasibility violation in our paper is better but the computational time grows much faster. This makes sense since we run the dual simplex algorithm, which iterates over the feasible solutions but is more computationally expensive than the interior-point algorithm. Figure 3 demonstrates that the structure of the FS-WBP is not favorable to the dual simplex algorithm, partially confirming our computational hardness results in Section 3.

| m | g | f2 |
|---|---|---|
| | normalized obj | |
| 200 | - | 3.6e-03±3.1e-04 |
| 500 | - | 4.4e-03±6.2e-04 |
| 1000 | - | 4.8e-03±5.4e-04 |
| 2000 | - | 5.0e-03±3.8e-04 |
| | feasibility | |
| 200 | 3.2e-07±1.8e-07 | 7.4e-07±1.8e-07 |
| 500 | 2.8e-07±5.0e-08 | 7.0e-07±2.8e-07 |
| 1000 | 2.1e-07±1.0e-07 | 7.1e-07±2.0e-07 |
| 2000 | 2.0e-07±1.3e-07 | 8.7e-07±2.0e-07 |
| | iteration | |
| 200 | 1.5e+05±2.4e+04 | 2.4e+03±3.2e+02 |
| 500 | 5.0e+05±8.8e+04 | 3.3e+03±1.4e+03 |
| 1000 | 1.3e+06±1.5e+05 | 1.9e+03±3.1e+02 |
| 2000 | 4.9e+06±1.6e+06 | 4.5e+03±1.7e+03 |

Figure 3: Preliminary results with Gurobi and the FASTIBP algorithm ($\eta = 0.001$).

**MNIST images.** We apply the FASTIBP algorithm ($\eta = 0.001$) to compute the Wasserstein barycenter of the resulting images for each digit on the MNIST dataset and compare it to IBP ($\eta = 0.001$). We exclude BADMM since Yang et al. [56, Figure 3] and Ge et al. [26, Table 1] have shown that IBP outperforms BADMM on the MNIST dataset. The approximate barycenters obtained by the FASTIBP and IBP algorithms are presented in Table 1. It can be seen that the FASTIBP algorithm provides a "sharper" approximate barycenter than IBP when $\eta = 0.001$ is set for both. This demonstrates the good quality of the solution obtained by our algorithm.

## 6 Conclusions

In this paper, we prove that the fixed-support Wasserstein barycenter problem (FS-WBP) in the standard LP form is not a minimum-cost flow (MCF) problem when $m \geq 3$ and $n \geq 3$. Thus, the direct application of network flow algorithms to the FS-WBP in standard LP form is inefficient, supporting the favorable performance of various algorithms based on other reformulation of the FS-WBP. We also propose a *deterministic* variant of iterative Bregman projection (IBP) algorithm, namely FASTIBP, with the complexity bound of $\widetilde{O}(mn^{7/3}\varepsilon^{-4/3})$. This bound is better than that of $\widetilde{O}(mn^2\varepsilon^{-2})$ from the IBP algorithm in terms of $\varepsilon$, and that of $\widetilde{O}(mn^{5/2}\varepsilon^{-1})$ from other accelerated algorithms in terms of $n$. Experiments on synthetic data and real images demonstrate the favorable performance of the FASTIBP algorithm in practice.

## Broader Impact

The problem of computing barycenter of probability measures has become increasingly important in several application domains, including physics, economics, machine learning, statistics, and data science. However, in these applications, the number of supports of the probability measures, such as images, can be very large. In this work, we study the fundamental hardness of the problem and propose efficient and scalable algorithm to solve for the fixed-support barycenters. Our work provides a new deterministic algorithm for computer scientists, physicists, economists, and statistician to tackle computationally expensive problems in their application domains and potentially accelerate scientific discoveries. We do not foresee any negative impact to society from our work.

## Acknowledgements

We would like to thank the area chair and four anonymous referees for constructive suggestions that improve the paper. Xi Chen is supported by National Science Foundation via the Grant IIS-1845444. This work is supported in part by the Mathematical Data Science program of the Office of Naval Research under grant number N00014-18-1-2764.

## Footnotes

[1]We implement ADMM [56], APDAGD [32] and accelerated IBP [29] and find that they perform worse than our algorithm. However, we believe it is largely due to our own implementation issue since these algorithms require fine hyper-parameter tuning. We are also unaware of any public codes available online. Thus, we exclude them for a fair comparison.

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
