[Supplementary Material]

# A Dual Entropic Regularized FS-WBP

In this section, we provide a complete derivation of the problem in Eq. (6).

By introducing the dual variables $\{\alpha_1, \alpha_2, \ldots, \alpha_m, \beta_1, \beta_2, \ldots, \beta_{m-1}\} \subseteq \mathbb{R}^n$, we define the Lagrangian function of the entropic regularized FS-WBP in Eq. (4) as follows:

$$\mathcal{L}(X_1, \ldots, X_m, \alpha_1, \ldots, \alpha_m, \beta_1, \ldots, \beta_{m-1}) \tag{12}$$
$$= \sum_{k=1}^{m} \omega_k (\langle C_k, X_k \rangle - \eta H(X_k)) - \sum_{k=1}^{m} \alpha_k^\top (r(X_k) - u^k) - \sum_{k=1}^{m-1} \beta_k^\top (c(X_{k+1}) - c(X_k)).$$

By the definition of $H(X)$, the nonnegative constraint $X \geq 0$ can be neglected. In order to derive the smooth dual objective function, we consider the following minimization problem:

$$\min_{\{(X_1, \ldots, X_m): \|X_k\|_1 = 1, \forall k \in [m]\}} \mathcal{L}(X_1, \ldots, X_m, \alpha_1, \ldots, \alpha_m, \beta_1, \ldots, \beta_{m-1}).$$

In the above problem, the objective function is strongly convex. Thus, the optimal solution is unique. For the simplicity, we let $\alpha = (\alpha_1, \alpha_2, \ldots, \alpha_m) \in \mathbb{R}^{mn}$ and $\beta = (\beta_1, \beta_2, \ldots, \beta_{m-1}) \in \mathbb{R}^{(m-1)n}$ and assume the convention $\beta_0 = \beta_m = \mathbf{0}_n$. After the simple calculations, the optimal solution $\bar{X}(\alpha, \beta) = (\bar{X}_1(\alpha, \beta), \ldots, \bar{X}_m(\alpha, \beta))$ has the following form:

$$(\bar{X}_k(\alpha, \beta))_{ij} = \frac{e^{\eta^{-1}(\omega_k^{-1}(\alpha_{ki} + \beta_{k-1,j} - \beta_{kj}) - (C_k)_{ij})}}{\sum_{1 \leq i,j \leq n} e^{\eta^{-1}(\omega_k^{-1}(\alpha_{ki} + \beta_{k-1,j} - \beta_{kj}) - (C_k)_{ij})}} \quad \text{for all } k \in [m], \tag{13}$$

Plugging Eq. (13) into Eq. (12) yields that the dual form is:

$$\max_{\alpha_1, \ldots, \alpha_m, \beta_1, \ldots, \beta_{m-1}} \left\{ -\eta \sum_{k=1}^{m} \omega_k \log \left( \sum_{1 \leq i,j \leq n} e^{\eta^{-1}(\omega_k^{-1}(\alpha_{ki} + \beta_{k-1,j} - \beta_{kj}) - (C_k)_{ij})} \right) + \sum_{k=1}^{m} \alpha_k^\top u_k \right\}.$$

In order to streamline our subsequent presentation, we perform a change of variables $\lambda_k = (\eta\omega_k)^{-1}\alpha_k$ and $\tau_k = (\eta\omega_k)^{-1}(\beta_{k-1} - \beta_k)$ for all $k \in [m]$. Recall that $\beta_0 = \beta_m = \mathbf{0}_n$, we have $\sum_{k=1}^{m} \omega_k \tau_k = \mathbf{0}_n$. Putting these pieces together, we reformulate the problem as

$$\min_{\lambda, \tau \in \mathbb{R}^{mn}} \varphi(\lambda, \tau) := \sum_{k=1}^{m} \omega_k \log \left( \sum_{1 \leq i,j \leq n} e^{\lambda_{ki} + \tau_{kj} - \eta^{-1}(C_k)_{ij}} \right) - \sum_{k=1}^{m} \omega_k \lambda_k^\top u^k, \quad \text{s.t. } \sum_{k=1}^{m} \omega_k \tau_k = \mathbf{0}_n.$$

To further simplify the above formulation, we use the notation $B_k(\lambda, \tau) \in \mathbb{R}^{n \times n}$ by $(B_k(\lambda_k, \tau_k))_{ij} = e^{\lambda_{ki} + \tau_{kj} - \eta^{-1}(C_k)_{ij}}$ for all $(i, j) \in [n] \times [n]$. To this end, we obtain the dual entropic-regularized FS-WBP problem defined by

$$\min_{\lambda, \tau \in \mathbb{R}^{mn}} \varphi(\lambda, \tau) := \sum_{k=1}^{m} \omega_k \left( \log(\|B_k(\lambda_k, \tau_k)\|_1) - \lambda_k^\top u^k \right), \quad \text{s.t. } \sum_{k=1}^{m} \omega_k \tau_k = \mathbf{0}_n.$$

# B Minimal-Cost Flow Problem

We specify the definition of the minimum-cost flow (MCF) problem.

**Definition B.1.** *The MCF problem finds the cheapest possible way of sending a certain amount of flow through a flow network. Formally,*

$$
\begin{array}{ll}
\min & \sum_{(u,v) \in E} f(u,v) \cdot a(u,v) \\
\text{s.t.} & f(u,v) \geq 0, \quad \text{for all } (u,v) \in E, \\
& f(u,v) \leq c(u,v) \quad \text{for all } (u,v) \in E, \\
& f(u,v) = -f(v,u) \quad \text{for all } (u,v) \in E, \\
& \sum_{(u,w) \in E \text{ or } (w,u) \in E} f(u,w) = 0, \\
& \sum_{w \in V} f(s,w) = d \quad \text{and} \quad \sum_{w \in V} f(w,t) = d.
\end{array}
\tag{14}
$$

*The flow network $G = (V, E)$ is a directed graph $G = (V, E)$ with a source vertex $s \in V$ and a sink vertex $t \in V$, where each edge $(u, v) \in E$ has capacity $c(u, v) > 0$, flow $f(u, v) \geq 0$ and cost $a(u, v)$, with most MCF algorithms supporting edges with negative costs. The cost of sending this flow along an edge $(u, v)$ is $f(u, v) \cdot a(u, v)$. The problem requires an amount of flow $d$ to be sent from source $s$ to sink $t$. Our goal is to minimize the total cost of the flow over all edges.*

## C  An Illustrative Example

We provide an illustrative counterexample for showing that the FS-WBP in Eq. (10) is not an MCF problem when $m = 3$ and $n = 3$.

**Example C.1.** *When $m = 3$ and $n = 3$, the constraint matrix is*

$$A = \begin{pmatrix} -I_3 \otimes \mathbf{1}_3^\top & \mathbf{0}_{3\times9} & \mathbf{0}_{3\times9} \\ \mathbf{0}_{3\times9} & I_3 \otimes \mathbf{1}_3^\top & \mathbf{0}_{3\times9} \\ \mathbf{0}_{3\times9} & \mathbf{0}_{3\times9} & -I_3 \otimes \mathbf{1}_3^\top \\ \mathbf{1}_3^\top \otimes I_3 & -\mathbf{1}_3^\top \otimes I_3 & \mathbf{0}_{3\times9} \\ \mathbf{0}_{3\times9} & -\mathbf{1}_3^\top \otimes I_3 & \mathbf{1}_3^\top \otimes I_3 \end{pmatrix}.$$

*Setting the set $I = \{1, 4, 7, 10, 11, 13, 15\}$ and letting $e_1$, $e_2$ and $e_3$ be the first, second and third standard basis row vectors in $\mathbb{R}^n$, the resulting matrix with the rows in $I$ is*

$$R = \begin{pmatrix} -e_1 \otimes \mathbf{1}_3^\top & \mathbf{0}_{1\times9} & \mathbf{0}_{1\times9} \\ \mathbf{0}_{1\times9} & e_1 \otimes \mathbf{1}_3^\top & \mathbf{0}_{1\times9} \\ \mathbf{0}_{1\times9} & \mathbf{0}_{1\times9} & -e_1 \otimes \mathbf{1}_3^\top \\ \mathbf{1}_3^\top \otimes e_1 & -\mathbf{1}_3^\top \otimes e_1 & \mathbf{0}_{1\times9} \\ \mathbf{1}_3^\top \otimes e_2 & -\mathbf{1}_3^\top \otimes e_2 & \mathbf{0}_{1\times9} \\ \mathbf{0}_{1\times9} & -\mathbf{1}_3^\top \otimes e_1 & \mathbf{1}_3^\top \otimes e_1 \\ \mathbf{0}_{1\times9} & -\mathbf{1}_3^\top \otimes e_3 & \mathbf{1}_3^\top \otimes e_3 \end{pmatrix}.$$

*Instead of considering all columns of $R$, it suffices to show that no partition of $I$ guarantees for any $j \in \{1, 2, 11, 12, 13, 19, 21\}$ that*

$$\sum_{i \in I_1} R_{ij} - \sum_{i \in I_2} R_{ij} \in \{-1, 0, 1\}.$$

*We write the submatrix of $R$ with these columns as*

$$\bar{R} = \begin{pmatrix} -1 & -1 & 0 & 0 & 0 & 0 & 0 \\ 0 & 0 & 1 & 1 & 0 & 0 & 0 \\ 0 & 0 & 0 & 0 & 0 & -1 & -1 \\ 1 & 0 & 0 & 0 & -1 & 0 & 0 \\ 0 & 1 & -1 & 0 & 0 & 0 & 0 \\ 0 & 0 & 0 & 0 & -1 & 1 & 0 \\ 0 & 0 & 0 & -1 & 0 & 0 & 1 \end{pmatrix}$$

*First, we claim that rows 1, 2, 4, 5 and 7 are in the same set $I_1$. Indeed, columns 1 and 2 imply that rows 1, 4 and 5 are in the same set. Column 3 and 4 imply that rows 2, 5 and 7 are in the same set. Putting these pieces together yields the desired claim. Then we consider the set that the row 6 belongs to and claim a contradiction. Indeed, row 6 can not be in $I_1$ since column 5 implies that rows 4 and 6 are not in the same set. However, row 6 must be in $I_1$ since columns 6 and 7 imply that rows 3, 6 and 7 are in the same set. Putting these pieces yields the contradiction. Thus, by using Propositions 3.1 and 3.3, $A$ is not TU and problem (10) is not a MCF problem when $m = 3$ and $n = 3$.*

## D  Missing Proofs in Section 2 and 3

In this section, we present the missing proofs of Lemma 2.2, Lemma 2.5, Proposition 3.3, Theorem 3.4 and Theorem 3.5.

### D.1  Proof of Lemma 2.2

First, we show that there exists $m$ pairs of optimal solutions $\{(\lambda^j, \tau^j)\}_{j \in [m]}$ such that each of $(\lambda^j, \tau^j)$ satisfies that

$$\max_{1 \le i \le n} (\tau_k^j)_i \ge 0 \ge \min_{1 \le i \le n} (\tau_k^j)_i, \quad \text{for all } k \ne j. \tag{15}$$

Fixing $j \in [m]$, we let $(\widehat{\lambda}, \widehat{\tau})$ be an optimal solution of the dual entropic regularized FS-WBP in Eq. (6). If $\widehat{\tau}$ satisfies Eq. (15), the claim holds true for the fixed $j \in [m]$. Otherwise, we define $m-1$ shift terms given by

$$\Delta\widehat{\tau}_k = \frac{\max_{1 \leq i \leq n}(\widehat{\tau}_k)_i + \min_{1 \leq i \leq n}(\widehat{\tau}_k)_i}{2} \in \mathbb{R} \quad \text{for all } k \neq j,$$

and let $(\lambda^j, \tau^j)$ with

$$\begin{aligned}
\tau_k^j &= \widehat{\tau}_k - \Delta\widehat{\tau}_k \mathbf{1}_n, & \lambda_k^j &= \widehat{\lambda}_k + \Delta\widehat{\tau}_k \mathbf{1}_n, & \text{for all } k \neq j, \\
\tau_j^j &= \widehat{\tau}_j + (\textstyle\sum_{k \neq j}(\frac{\omega_k}{\omega_j}))\Delta\widehat{\tau}_k \mathbf{1}_n, & \lambda_j^j &= \widehat{\lambda}_j - (\textstyle\sum_{k \neq j}(\frac{\omega_k}{\omega_j}))\Delta\widehat{\tau}_k \mathbf{1}_n.
\end{aligned}$$

Using this construction, we have $(\lambda_k^j)_i + (\tau_k^j)_{i'} = (\widehat{\lambda}_k)_i + (\widehat{\tau}_k)_{i'}$ for all $i, i' \in [n]$ and all $k \in [m]$. This implies that $B_k(\widehat{\lambda}_k, \widehat{\tau}_k) = B_k(\lambda_k^{k'}, \tau_k^{k'})$ for all $k \in [m]$. Furthermore, we have

$$\sum_{k=1}^m \omega_k \tau_k^j = \sum_{k=1}^m \omega_k \widehat{\tau}_k, \qquad \sum_{k=1}^m \omega_k (\lambda_k^j)^\top u^k = \sum_{k=1}^m \omega_k \widehat{\lambda}_k^\top u^k.$$

Putting these pieces together yields $\varphi(\lambda^j, \tau^j) = \varphi(\widehat{\lambda}, \widehat{\tau})$. Moreover, by the definition of $(\lambda^j, \tau^j)$ and $m-1$ shift terms, $\tau^j$ satisfies Eq. (15). Therefore, we conclude that $(\lambda^j, \tau^j)$ is an optimal solution that satisfies Eq. (15) for the fixed $j \in [m]$. Since $j \in [m]$ is chosen arbitarily, we can find the desired pairs of optimal solutions $\{(\lambda^j, \tau^j)\}_{j \in [m]}$ satisfying Eq. (15) by repeating the above argument $m$ times.

Furthermore, each of $(\lambda^j, \tau^j)$ must satisfy the optimality condition for Eq. (6) for all $k \in [m]$. Fixing $j \in [m]$, there exists $z \in \mathbb{R}^n$ such that

$$\sum_{k=1}^m \omega_k \tau_k^j = \mathbf{0}_n \quad \text{and} \quad \frac{B_k(\lambda_k^j, \tau_k^j)^\top \mathbf{1}_n}{\|B_k(\lambda_k^j, \tau_k^j)\|_1} - z = \mathbf{0}_n \quad \text{for all } k \in [m]. \tag{16}$$

By the definition of $B_k(\cdot, \cdot)$, we have

$$\tau_k^j = \log(z) + \log(\|B_k(\lambda_k^j, \tau_k^j)\|_1)\mathbf{1}_n - \log(e^{-\eta^{-1}C_k}\mathrm{diag}(e^{\lambda_k^j})\mathbf{1}_n) \text{ for all } k \in [m].$$

This together with the first equality in Eq. (16) yields that

$$\tau_k^j = \sum_{l=1}^m \omega_l \log(e^{-\eta^{-1}C_l}\mathrm{diag}(e^{\lambda_l^j})\mathbf{1}_n) - \log(e^{-\eta^{-1}C_k}\mathrm{diag}(e^{\lambda_k^j})\mathbf{1}_n) \text{ for all } k \in [m].$$

For each $i \in [n]$ and $l \in [m]$, by the nonnegativity of each entry of $C_l$, we have

$$-\eta^{-1}\|C_l\|_\infty + \log(\mathbf{1}_n^\top e^{\lambda_l^j}) \leq [\log(e^{-\eta^{-1}C_l}\mathrm{diag}(e^{\lambda_l^j})\mathbf{1}_n)]_i \leq \log(\mathbf{1}_n^\top e^{\lambda_l^j}).$$

Putting these pieces together yields

$$\max_{1 \leq i \leq n}(\tau_k^j)_i - \min_{1 \leq i \leq n}(\tau_k^j)_i \leq \eta^{-1}\|C_k\|_\infty + \sum_{l=1}^m \omega_l \eta^{-1}\|C_l\|_\infty \text{ for all } k \in [m]. \tag{17}$$

Combining Eq. (15) and Eq. (17) yields that

$$\|\tau_k^j\|_\infty \leq \eta^{-1}\|C_k\|_\infty + \sum_{l=1}^m \omega_l \eta^{-1}\|C_l\|_\infty \text{ for all } k \neq j. \tag{18}$$

Since $\sum_{k=1}^m \omega_k \tau_k^j = \mathbf{0}_n$, we have

$$\|\tau_j^j\|_\infty \leq \omega_j^{-1}\sum_{k \neq j} \omega_k \|\tau_k^j\|_\infty \leq (\eta\omega_j)^{-1}\sum_{k \neq j} \omega_k \|C_k\|_\infty + (\eta\omega_j)^{-1}(1 - \omega_j)\sum_{k=1}^m \omega_k \|C_k\|_\infty.$$

Finally, we proceed to the key part and define the averaging iterate

$$\lambda^\star = \sum_{j=1}^m \omega_j \lambda^j, \qquad \tau^\star = \sum_{j=1}^m \omega_j \tau^j.$$

Since $\varphi$ is convex and $(\omega_1, \omega_2, \ldots, \omega_m) \in \Delta^m$, we have $\varphi(\lambda^\star, \tau^\star) \leq \sum_{j=1}^m \omega_j \varphi(\lambda^j, \tau^j)$ and $\sum_{k=1}^m \omega_k \tau_k^\star = \mathbf{0}_n$. Since $(\lambda^j, \tau^j)$ are optimal solutions for all $j \in [m]$, we conclude that $(\lambda^\star, \tau^\star)$ is an optimal solution.

The remaining step is to show that $\|\lambda_k^\star\|_\infty \leq R_\lambda$ and $\|\tau_k^\star\|_\infty \leq R_\tau$ for all $k \in [m]$. More specifically, we have

$$
\begin{aligned}
\|\tau_k^\star\|_\infty &\leq \sum_{j=1}^m \omega_j \|\tau_k^j\|_\infty = \omega_k \|\tau_k^k\|_\infty + \sum_{j \neq k} \omega_j \|\tau_k^j\|_\infty \\
&\leq \eta^{-1} \sum_{l \neq k} \omega_l \|C_l\|_\infty + \eta^{-1}(1 - \omega_k) \sum_{l=1}^m \omega_l \|C_l\|_\infty + \eta^{-1}(1 - \omega_k)(\|C_k\|_\infty + \sum_{l=1}^m \omega_l \|C_l\|_\infty) \\
&\leq \eta^{-1} \|C_k\|_\infty + 3\eta^{-1} \sum_{l=1}^m \omega_l \|C_l\|_\infty \\
&\leq 4\eta^{-1} \bar{C} = R_\tau.
\end{aligned}
$$

Since $(\lambda^\star, \tau^\star)$ is an optimal solution, it satisfies the optimality condition for Eq. (6). Formally, we have

$$
\frac{B_k(\lambda_k^\star, \tau_k^\star)\mathbf{1}_n}{\|B_k(\lambda_k^\star, \tau_k^\star)\|_1} - u^k = \mathbf{0}_n \quad \text{for all } k \in [m]. \tag{19}
$$

By the definition of $B_k(\cdot, \cdot)$, we have

$$
\lambda_k^\star = \log(u^k) + \log(\|B_k(\lambda_k^\star, \tau_k^\star)\|_1)\mathbf{1}_n - \log(e^{-\eta^{-1}C_k}\mathrm{diag}(e^{\tau_k^\star})\mathbf{1}_n) \text{ for all } k \in [m].
$$

This implies that

$$
\begin{aligned}
\max_{1 \leq i \leq n} (\lambda_k^\star)_i &\leq \eta^{-1}\|C_k\|_\infty + \log(n) + \|\tau_k^\star\|_\infty + \log(\|B_k(\lambda_k^\star, \tau_k^\star)\|_1), \\
\min_{1 \leq i \leq n} (\lambda_k^\star)_i &\geq \log(\bar{u}) - \log(n) - \|\tau_k^\star\|_\infty + \log(\|B_k(\lambda_k^\star, \tau_k^\star)\|_1).
\end{aligned}
$$

Therefore, we conclude that

$$
\|\lambda_k^\star\|_\infty \leq \eta^{-1}\|C_k\|_\infty + \log(n) - \log(\bar{u}) + \|\tau_k^\star\|_\infty.
$$

This completes the proof.

### D.2 Proof of Lemma 2.5

The first statement directly follows from the definition of $\varphi$ in Eq. (6). For the second statement, we provide the explicit form of the gradient of $\varphi_k$ as follows,

$$
\nabla\varphi_k(\lambda, \tau) = \begin{pmatrix} \frac{B_k(\lambda, \tau)\mathbf{1}_n}{\mathbf{1}_n^\top B_k(\lambda, \tau)\mathbf{1}_n} - u^k \\ \frac{B_k(\lambda, \tau)^\top \mathbf{1}_n}{\mathbf{1}_n^\top B_k(\lambda, \tau)\mathbf{1}_n} \end{pmatrix}.
$$

Now we construct the following entropic regularized OT problem,

$$
\min_{X:\|X\|_1=1} \langle C_k, X \rangle - \eta H(X), \quad \text{s.t.} X\mathbf{1}_n = (1/n)\mathbf{1}_n, \ X^\top\mathbf{1}_n = (1/n)\mathbf{1}_n.
$$

Since the function $-H(X)$ is strongly convex with respect to the $\ell_1$-norm on the probability simplex $Q \subseteq \mathbb{R}^{n^2}$, the above entropic regularized OT problem is a special case of the following linearly constrained convex optimization problem:

$$
\min_{x \in Q} f(x), \quad \text{s.t. } Ax = b,
$$

where $f$ is strongly convex with respect to the $\ell_1$-norm on the set $Q$. We use the $\ell_2$-norm for the dual space of the Lagrange multipliers. By Nesterov [40, Theorem 1] and the fact that $\|A\|_{1 \to 2} = 2$, the dual objective function $\tilde{\varphi}_k$ satisfies the following inequality:

$$
\|\nabla\tilde{\varphi}_k(\alpha, \beta) - \nabla\tilde{\varphi}_k(\alpha', \beta')\| \leq \frac{4}{\eta} \left\| \begin{pmatrix} \alpha \\ \beta \end{pmatrix} - \begin{pmatrix} \alpha' \\ \beta' \end{pmatrix} \right\|.
$$

Recall that the function $\tilde{\varphi}_k$ is given by

$$\tilde{\varphi}_k(\alpha, \beta) = \eta \log \left( \sum_{i,j=1}^{n} e^{-\frac{(C_k)_{ij} - \alpha_i - \beta_j}{\eta} - 1} \right) - \langle \alpha, u \rangle - \langle \beta, v \rangle + \eta.$$

This together with the definition of $B_k(\cdot, \cdot)$ implies that

$$\nabla \tilde{\varphi}_k(\alpha, \beta) = \begin{pmatrix} \frac{B_k\left(\eta^{-1}\alpha - (1/2)\mathbf{1}_n, \eta^{-1}\beta - (1/2)\mathbf{1}_n\right)\mathbf{1}_n}{\mathbf{1}_n^\top B_k(\eta^{-1}\alpha - (1/2)\mathbf{1}_n, \eta^{-1}\beta - (1/2)\mathbf{1}_n)\mathbf{1}_n} - u \\ \frac{B_k\left(\eta^{-1}\alpha - (1/2)\mathbf{1}_n, \eta^{-1}\beta - (1/2)\mathbf{1}_n\right)^\top \mathbf{1}_n}{\mathbf{1}_n^\top B_k(\eta^{-1}\alpha - (1/2)\mathbf{1}_n, \eta^{-1}\beta - (1/2)\mathbf{1}_n)\mathbf{1}_n} - v \end{pmatrix}$$

Performing the change of variable $\lambda = \eta^{-1}\alpha - (1/2)\mathbf{1}_n$ and $\tau = \eta^{-1}\beta - (1/2)\mathbf{1}_n$ (resp. $\lambda'$ and $\tau'$), we have

$$
\begin{aligned}
& \|\nabla \varphi_k(\lambda, \tau) - \nabla \varphi_k(\lambda', \tau')\| \\
= \; & \|\nabla \tilde{\varphi}_k(\eta(\lambda + (1/2)\mathbf{1}_n), \eta(\tau + (1/2)\mathbf{1}_n)) - \nabla \tilde{\varphi}_k(\eta(\lambda' + (1/2)\mathbf{1}_n), \eta(\tau' + (1/2)\mathbf{1}_n))\| \\
\leq \; & 4 \left\| \begin{pmatrix} \lambda' - \lambda \\ \tau' - \tau \end{pmatrix} \right\|.
\end{aligned}
$$

This completes the proof.

### D.3   Proof of Proposition 3.3

The standard LP representation of the MCF problem is

$$\min_{x \in \mathbb{R}^{|E|}} c^\top x, \quad \text{s.t. } Ax = b, \; l \leq x \leq u.$$

where $x \in \mathbb{R}^{|E|}$ with $x_j$ being the flow through arc $j$, $b \in \mathbb{R}^{|V|}$ with $b_i$ being external supply at node $i$ and $\mathbf{1}^\top b = 0$, $c_j$ is unit cost of flow through arc $j$, $l_j$ and $u_j$ are lower and upper bounds on flow through arc $j$ and $A \in \mathbb{R}^{|V| \times |E|}$ is the arc-node incidence matrix with entries

$$A_{ij} = \begin{cases} -1 & \text{if arc } j \text{ starts at node } i \\ 1 & \text{if arc } j \text{ ends at node } i \\ 0 & \text{otherwise} \end{cases}.$$

Since each arc has two endpoints, the constraint matrix $A$ is a $\{-1, 0, 1\}$-valued matrix in which each column contains two nonzero entries 1 and $-1$. Using Proposition 3.2, we obtain that $A$ is TU and the rows of $A$ are categorized into a single set.

### D.4   Proof of Theorem 3.4

When $n = 2$, the constraint matrix $A$ has $E = I_2 \otimes \mathbf{1}_2^\top$ and $G = \mathbf{1}_2^\top \otimes I_2$. The matrix $A \in \mathbb{R}^{(4m-2) \times 4m}$ is a $\{-1, 0, 1\}$-valued matrix with several redundant rows and each column has at most three nonzero entries in $\{-1, 0, 1\}$. Now we simplify the matrix $A$ by removing a specific set of redundant rows. In particular, we observe that

$$\sum_{i \in \{1,2,3,4,2m+1,2m+2\}} a_{ij} = 0, \quad \forall j \in [4m],$$

which implies that the $(2m + 2)$th row is redundant. Similarly, we have

$$\sum_{i \in \{3,4,5,6,2m+3,2m+4\}} a_{ij} = 0, \quad \forall j \in [4m],$$

which implies that the $(2m + 3)$th row is redundant. Using this argument, we remove $m - 1$ rows from the last $2m - 2$ rows. The resulting matrix $\bar{A} \in \mathbb{R}^{(3m-1) \times 4m}$ has very nice structure such that

each column has only two nonzero entries $1$ and $-1$; see the following matrix when $m$ is odd:

$$\bar{A} = \left(\begin{array}{ccccc}
-E & \cdots & \cdots & \cdots & \cdots \\
\vdots & E & \ddots & \ddots & \vdots \\
\vdots & \ddots & \ddots & \ddots & \vdots \\
\vdots & \ddots & \ddots & (-1)^{m-1}E & \vdots \\
\vdots & \ddots & \ddots & \ddots & (-1)^{m}E \\
\hdashline
\mathbf{1}_2^\top \otimes e_1 & -\mathbf{1}_2^\top \otimes e_1 & \ddots & \ddots & \vdots \\
\vdots & -\mathbf{1}_2^\top \otimes e_2 & \mathbf{1}_2^\top \otimes e_2 & \ddots & \vdots \\
\vdots & \ddots & \ddots & \ddots & \vdots \\
\cdots & \cdots & \cdots & (-1)^{m}\mathbf{1}_2^\top \otimes e_2 & (-1)^{m+1}\mathbf{1}_2^\top \otimes e_2
\end{array}\right).$$

where $e_1$ and $e_2$ are respectively the first and second standard basis (row) vectors in $\mathbb{R}^2$. Furthermore, the rows of $\bar{A}$ are categorized into a single set so that the criterion in Proposition 3.2 holds true (the dashed line in the formulation of $\bar{A}$ serves as a partition of this single set into two sets). Using Proposition 3.2, we conclude that $\bar{A}$ is TU.

## D.5   Proof of Theorem 3.4

We use the proof by contradiction. In particular, assume that problem (10) is a MCF problem when $m \geq 3$ and $n \geq 3$, Proposition 3.3 implies that the constraint matrix $A$ is TU. Since $A$ is a $\{-1, 0, 1\}$-valued matrix, Proposition 3.1 further implies that for each set $I \subseteq [2mn - n]$ there is a partition $I_1$, $I_2$ of $I$ such that

$$\sum_{i \in I_1} a_{ij} - \sum_{i \in I_2} a_{ij} \in \{-1, 0, 1\}, \quad \forall j \in [mn^2]. \tag{20}$$

In what follows, for any given $m \geq 3$ and $n \geq 3$, we construct a set of rows $I$ such that no partition of $I$ guarantees that Eq. (20) holds true. For the ease of presentation, we rewrite the matrix $A \in \mathbb{R}^{(2mn-n) \times mn^2}$ as follows,

$$A = \left(\begin{array}{ccccc}
-I_n \otimes \mathbf{1}_n^\top & \cdots & \cdots & \cdots & \cdots \\
\vdots & I_n \otimes \mathbf{1}_n^\top & \ddots & \ddots & \vdots \\
\vdots & \ddots & \ddots & \ddots & \vdots \\
\vdots & \ddots & \ddots & (-1)^{m-1}I_n \otimes \mathbf{1}_n^\top & \vdots \\
\vdots & \ddots & \ddots & \ddots & (-1)^{m}I_n \otimes \mathbf{1}_n^\top \\
\hdashline
\mathbf{1}_n^\top \otimes I_n & -\mathbf{1}_n^\top \otimes I_n & \ddots & \ddots & \vdots \\
\vdots & -\mathbf{1}_n^\top \otimes I_n & \mathbf{1}_n^\top \otimes I_n & \ddots & \vdots \\
\vdots & \ddots & \ddots & \ddots & \vdots \\
\cdots & \cdots & \cdots & (-1)^{m}\mathbf{1}_n^\top \otimes I_n & (-1)^{m+1}\mathbf{1}_n^\top \otimes I_n
\end{array}\right).$$

Setting the set $I = \{1, n+1, 2n+1, 3n+1, 3n+2, 4n+1, 4n+3\}$ and letting $e_1$, $e_2$ and $e_3$ be the first, second and third standard basis row vectors in $\mathbb{R}^n$, the resulting matrix with the rows in $I$ is

$$R = \left(\begin{array}{cccccc}
-e_1 \otimes \mathbf{1}_n^\top & \mathbf{0}_{1 \times n^2} & \mathbf{0}_{1 \times n^2} & \mathbf{0}_{1 \times n^2} & \cdots & \mathbf{0}_{1 \times n^2} \\
\mathbf{0}_{1 \times n^2} & e_1 \otimes \mathbf{1}_n^\top & \mathbf{0}_{1 \times n^2} & \mathbf{0}_{1 \times n^2} & \cdots & \mathbf{0}_{1 \times n^2} \\
\mathbf{0}_{1 \times n^2} & \mathbf{0}_{1 \times n^2} & -e_1 \otimes \mathbf{1}_n^\top & \mathbf{0}_{1 \times n^2} & \cdots & \mathbf{0}_{1 \times n^2} \\
\hdashline
\mathbf{1}_n^\top \otimes e_1 & -\mathbf{1}_n^\top \otimes e_1 & \mathbf{0}_{1 \times n^2} & \mathbf{0}_{1 \times n^2} & \cdots & \mathbf{0}_{1 \times n^2} \\
\mathbf{1}_n^\top \otimes e_2 & -\mathbf{1}_n^\top \otimes e_2 & \mathbf{0}_{1 \times n^2} & \mathbf{0}_{1 \times n^2} & \cdots & \mathbf{0}_{1 \times n^2} \\
\mathbf{0}_{1 \times n^2} & -\mathbf{1}_n^\top \otimes e_1 & \mathbf{1}_n^\top \otimes e_1 & \mathbf{0}_{1 \times n^2} & \cdots & \mathbf{0}_{1 \times n^2} \\
\mathbf{0}_{1 \times n^2} & -\mathbf{1}_n^\top \otimes e_3 & \mathbf{1}_n^\top \otimes e_3 & \mathbf{0}_{1 \times n^2} & \cdots & \mathbf{0}_{1 \times n^2}
\end{array}\right).$$

Instead of considering all columns of $R$, it suffices to show that no partition of $I$ guarantees

$$\sum_{i \in I_1} R_{ij} - \sum_{i \in I_2} R_{ij} \in \{-1, 0, 1\},$$

for all $j \in \{1, 2, n^2 + 2, n^2 + 3, n^2 + n + 1, 2n^2 + 1, 2n^2 + 3\}$. We write the submatrix of $R$ with these columns as

$$\bar{R} = \begin{pmatrix} -1 & -1 & 0 & 0 & 0 & 0 & 0 \\ 0 & 0 & 1 & 1 & 0 & 0 & 0 \\ 0 & 0 & 0 & 0 & 0 & -1 & -1 \\ 1 & 0 & 0 & 0 & -1 & 0 & 0 \\ 0 & 1 & -1 & 0 & 0 & 0 & 0 \\ 0 & 0 & 0 & 0 & -1 & 1 & 0 \\ 0 & 0 & 0 & -1 & 0 & 0 & 1 \end{pmatrix}.$$

Applying the same argument used in Example C.1, we obtain from Propositions 3.1 and 3.3 that $A$ is not TU when $m \geq 3$ and $n \geq 3$, which is a contradiction. As a consequence, the conclusion of the theorem follows.

# E  Technical Lemmas and Missing Proofs in Section 4

We present several technical lemmas which are important to analyzing the FASTIBP algorithm. The first lemma provides the inductive formula and the upper bound for $\theta_t$.

**Lemma E.1.** *Let $\{\theta_t\}_{t \geq 0}$ be the iterates generated by the FASTIBP algorithm. Then we have $0 < \theta_t \leq 2/(t+2)$ and $\theta_t^{-2} = (1 - \theta_{t+1})\theta_{t+1}^{-2}$ for all $t \geq 0$.*

The second lemma shows that all the iterates generated by the FASTIBP algorithm are feasible to the dual entropic regularized FS-WBP for all $t \geq 1$.

**Lemma E.2.** *Let $\{(\check{\lambda}^t, \check{\tau}^t)\}_{t \geq 0}$, $\{(\tilde{\lambda}^t, \tilde{\tau}^t)\}_{t \geq 0}$, $\{(\bar{\lambda}^t, \bar{\tau}^t)\}_{t \geq 0}$, $\{(\widehat{\lambda}^t, \widehat{\tau}^t)\}_{t \geq 0}$, $\{(\acute{\lambda}^t, \acute{\tau}^t)\}_{t \geq 0}$, and $\{(\lambda^t, \tau^t)\}_{t \geq 0}$ be the iterates generated by the FASTIBP algorithm. Then, we have*

$$\sum_{k=1}^{m} \omega_k \check{\tau}_k^t = \sum_{k=1}^{m} \omega_k \tilde{\tau}_k^t = \sum_{k=1}^{m} \omega_k \bar{\tau}_k^t = \sum_{k=1}^{m} \omega_k \widehat{\tau}_k^t = \sum_{k=1}^{m} \omega_k \acute{\tau}_k^t = \sum_{k=1}^{m} \omega_k \grave{\tau}_k^t = \sum_{k=1}^{m} \omega_k \tau_k^t = \mathbf{0}_n \quad \text{for all } t \geq 0.$$

The third lemma shows that the iterates $\{\tau^t\}_{t \geq 0}$ generated by the FASTIBP algorithm satisfies the bounded difference property: $\max_{1 \leq i \leq n}(\tau_k^t)_i - \min_{1 \leq i \leq n}(\tau_k^t)_i \leq R_\tau/2$.

**Lemma E.3.** *Let $\{(\lambda^t, \tau^t)\}_{t \geq 0}$ be the iterates generated by the FASTIBP algorithm. Then the following statement holds true:*

$$\max_{1 \leq i \leq n}(\tau_k^t)_i - \min_{1 \leq i \leq n}(\tau_k^t)_i \leq R_\tau/2,$$

*where $R_\tau > 0$ is defined in Lemma 2.2.*

The final lemma presents a key descent inequality for the FASTIBP algorithm.

**Lemma E.4.** *Let $\{(\check{\lambda}^t, \check{\tau}^t)\}_{t \geq 0}$ be the iterates generated by the FASTIBP algorithm and let $(\lambda^\star, \tau^\star)$ be an optimal solution in Lemma 2.2. Then the following statement holds true:*

$$\varphi(\check{\lambda}^{t+1}, \check{\tau}^{t+1}) - (1-\theta_t)\varphi(\check{\lambda}^t, \check{\tau}^t) - \theta_t \varphi(\lambda^\star, \tau^\star) \leq 2\theta_t^2 \left( \sum_{k=1}^{m} \omega_k \left( \left\| \begin{pmatrix} \lambda_k^\star - \tilde{\lambda}_k^t \\ \tau_k^\star - \tilde{\tau}_k^t \end{pmatrix} \right\|^2 - \left\| \begin{pmatrix} \lambda_k^\star - \tilde{\lambda}_k^{t+1} \\ \tau_k^\star - \tilde{\tau}_k^{t+1} \end{pmatrix} \right\|^2 \right) \right).$$

Then we proceed to the proofs of these technical lemmas, Theorem 4.2 and Theorem 4.3.

## E.1  Proof of Lemma E.1

By the definition of $\theta_t$, we have

$$\left( \frac{\theta_{t+1}}{\theta_t} \right)^2 = \frac{1}{4} \left( \sqrt{\theta_t^2 + 4} - \theta_t \right)^2 = 1 + \frac{\theta_t}{2} \left( \theta_t - \sqrt{\theta_t^2 + 4} \right) = 1 - \theta_{t+1},$$

which implies the desired inductive formula and $\theta_t > 0$ for all $t \geq 0$. Then we proceed to prove that $0 < \theta_t \leq 2/(t+2)$ for all $t \geq 0$ using the induction. Indeed, the claim trivially holds when $t = 0$ as $\theta_0 = 1$. Assume that the hypothesis holds for $t \leq t_0$, i.e., $\theta_{t_0} \leq 2/(t_0 + 2)$, we have

$$\theta_{t_0+1} = \frac{2}{1 + \sqrt{1 + \frac{4}{\theta_{t_0}^2}}} \leq \frac{2}{t_0 + 3}.$$

This completes the proof of the lemma.

## E.2 Proof of Lemma E.2

We first verify Lemma E.2 when $t = 0$. Indeed,

$$\sum_{k=1}^{m} \omega_k \check{\tau}_k^0 = \sum_{k=1}^{m} \omega_k \tilde{\tau}_k^0 = \mathbf{0}_n.$$

By the definition, $\bar{\tau}^0$ is a convex combination of $\check{\tau}^0$ and $\tilde{\tau}^0$ and $\hat{\tau}^0$ is a linear combination of $\bar{\tau}^0$, $\tilde{\tau}^1$ and $\tilde{\tau}^0$. Thus, we have

$$\sum_{k=1}^{m} \omega_k \bar{\tau}_k^0 = \sum_{k=1}^{m} \omega_k \hat{\tau}_k^0 = \mathbf{0}_n.$$

This also implies that $\sum_{k=1}^{m} \omega_k \acute{\tau}_k^0 = \mathbf{0}_n$. Using the update formula for $\grave{\tau}^0$, $\tau^0$ and $\check{\tau}^1$, we have

$$\sum_{k=1}^{m} \omega_k \grave{\tau}_k^0 = \sum_{k=1}^{m} \omega_k \tau_k^0 = \sum_{k=1}^{m} \omega_k \check{\tau}_k^1 = \mathbf{0}_n.$$

Besides that, the update formula for $\tilde{\tau}^1$ implies $\sum_{k=1}^{m} \omega_k \tilde{\tau}_k^1 = \mathbf{0}_n$. Repeating this argument, we obtain the desired equality in the conclusion of Lemma E.2 for all $t \geq 0$.

## E.3 Proof of Lemma E.3

We observe that $\tau_k^t = \grave{\tau}_k^t$ for all $k \in [m]$. By the update formula for $\grave{\tau}_k^t$, we have

$$\grave{\tau}_k^t = \acute{\tau}_k^t + \sum_{i=1}^{m} \omega_i \log(c_i) - \log(c_k) = \sum_{i=1}^{m} \omega_i \log(e^{-\eta^{-1}C_i} \mathrm{diag}(e^{\acute{\lambda}_i^t}) \mathbf{1}_n) - \log(e^{-\eta^{-1}C_k} \mathrm{diag}(e^{\acute{\lambda}_k^t}) \mathbf{1}_n).$$

After the simple calculation, we have

$$-\eta^{-1}\|C_k\|_\infty + \mathbf{1}_n^\top e^{\acute{\lambda}_k^t} \leq \log(e^{-\eta^{-1}C_k} \mathrm{diag}(e^{\acute{\lambda}_k^t}) \mathbf{1}_n)]_j \leq \mathbf{1}_n^\top e^{\acute{\lambda}_k^t}.$$

Therefore, the following inequality holds true for all $k \in [m]$,

$$\max_{1 \leq i \leq n} (\tau_k^t)_i - \min_{1 \leq i \leq n} (\tau_k^t)_i \leq \eta^{-1}\|C_k\|_\infty + \eta^{-1} \left( \sum_{i=1}^{m} \omega_i \|C_i\|_\infty \right) = 2\eta^{-1}(\max_{1 \leq k \leq m} \|C_k\|_\infty).$$

This together with the definition of $R_\tau$ yields the desired inequality.

## E.4 Proof of Lemma E.4

Using Lemma 2.5 with $(\lambda', \tau') = (\hat{\lambda}^{t+1}, \hat{\tau}^{t+1})$ and $(\lambda, \tau) = (\bar{\lambda}^t, \bar{\tau}^t)$, we have

$$\varphi(\hat{\lambda}^{t+1}, \hat{\tau}^{t+1}) \leq \varphi(\bar{\lambda}^t, \bar{\tau}^t) + \theta_t \begin{pmatrix} \tilde{\lambda}^{t+1} - \tilde{\lambda}^t \\ \tilde{\tau}^{t+1} - \tilde{\tau}^t \end{pmatrix}^\top \nabla\varphi(\bar{\lambda}^t, \bar{\tau}^t) + 2\theta_t^2 \left( \sum_{k=1}^{m} \omega_k \left\| \begin{pmatrix} \tilde{\lambda}_k^{t+1} - \tilde{\lambda}_k^t \\ \tilde{\tau}_k^{t+1} - \tilde{\tau}_k^t \end{pmatrix} \right\|^2 \right).$$

After some simple calculations, we find that

$$\varphi(\bar{\lambda}^t, \bar{\tau}^t) = (1 - \theta_t)\varphi(\bar{\lambda}^t, \bar{\tau}^t) + \theta_t\varphi(\bar{\lambda}^t, \bar{\tau}^t),$$

$$\begin{pmatrix} \tilde{\lambda}^{t+1} - \tilde{\lambda}^t \\ \tilde{\tau}^{t+1} - \tilde{\tau}^t \end{pmatrix}^\top \nabla\varphi(\bar{\lambda}^t, \bar{\tau}^t) = -\begin{pmatrix} \tilde{\lambda}^t - \bar{\lambda}^t \\ \tilde{\tau}^t - \bar{\tau}^t \end{pmatrix}^\top \nabla\varphi(\bar{\lambda}^t, \bar{\tau}^t) + \begin{pmatrix} \tilde{\lambda}^{t+1} - \bar{\lambda}^t \\ \tilde{\tau}^{t+1} - \bar{\tau}^t \end{pmatrix}^\top \nabla\varphi(\bar{\lambda}^t, \bar{\tau}^t).$$

Putting these pieces together yields that

$$\varphi(\widehat{\lambda}^{t+1}, \widehat{\tau}^{t+1}) \leq \underbrace{(1 - \theta_t)\varphi(\bar{\lambda}^t, \bar{\tau}^t) - \theta_t \begin{pmatrix} \tilde{\lambda}^t - \bar{\lambda}^t \\ \tilde{\tau}^t - \bar{\tau}^t \end{pmatrix}^\top \nabla\varphi(\bar{\lambda}^t, \bar{\tau}^t)}_{\text{I}} \tag{21}$$

$$+ \theta_t \underbrace{\left( \varphi(\bar{\lambda}^t, \bar{\tau}^t) + \begin{pmatrix} \tilde{\lambda}^{t+1} - \bar{\lambda}^t \\ \tilde{\tau}^{t+1} - \bar{\tau}^t \end{pmatrix}^\top \nabla\varphi(\bar{\lambda}^t, \bar{\tau}^t) + 2\theta_t \left( \sum_{k=1}^{m} \omega_k \left\| \begin{pmatrix} \tilde{\lambda}^{t+1} - \tilde{\lambda}^t \\ \tilde{\tau}^{t+1} - \tilde{\tau}^t \end{pmatrix} \right\|^2 \right) \right)}_{\text{II}}.$$

For the term I in equation (21), we derive from the definition of $(\bar{\lambda}^t, \bar{\tau}^t)$ that

$$-\theta_t \begin{pmatrix} \tilde{\lambda}^t - \bar{\lambda}^t \\ \tilde{\tau}^t - \bar{\tau}^t \end{pmatrix} = \theta_t \begin{pmatrix} \bar{\lambda}^t \\ \bar{\tau}^t \end{pmatrix} + (1 - \theta_t) \begin{pmatrix} \check{\lambda}^t \\ \check{\tau}^t \end{pmatrix} - \begin{pmatrix} \bar{\lambda}^t \\ \bar{\tau}^t \end{pmatrix} = (1 - \theta_t) \begin{pmatrix} \check{\lambda}^t - \bar{\lambda}^t \\ \check{\tau}^t - \bar{\tau}^t \end{pmatrix}.$$

Using this equality and the convexity of $\varphi$, we have

$$\text{I} = (1 - \theta_t) \left( \varphi(\bar{\lambda}^t, \bar{\tau}^t) + \begin{pmatrix} \check{\lambda}^t - \bar{\lambda}^t \\ \check{\tau}^t - \bar{\tau}^t \end{pmatrix}^\top \nabla\varphi(\bar{\lambda}^t, \bar{\tau}^t) \right) \leq (1 - \theta_t)\varphi(\check{\lambda}^t, \check{\tau}^t). \tag{22}$$

For the term II in equation (21), the definition of $(\tilde{\lambda}^{t+1}, \tilde{\tau}^{t+1})$ implies that

$$\begin{pmatrix} \lambda - \tilde{\lambda}^{t+1} \\ \tau - \tilde{\tau}^{t+1} \end{pmatrix}^\top \left( \nabla\varphi(\bar{\lambda}^t, \bar{\tau}^t) + 4\theta_t \begin{pmatrix} \omega_1(\tilde{\lambda}_1^{t+1} - \tilde{\lambda}_1^t) \\ \vdots \\ \omega_m(\tilde{\lambda}_m^{t+1} - \tilde{\lambda}_m^t) \\ \omega_1(\tilde{\tau}_1^{t+1} - \tilde{\tau}_1^t) \\ \vdots \\ \omega_m(\tilde{\tau}_m^{t+1} - \tilde{\tau}_m^t) \end{pmatrix} \right) \geq 0, \quad \text{for all } (\lambda, \tau) \in \mathbb{R}^{mn} \times \mathcal{P}.$$

Letting $(\lambda, \tau) = (\lambda^\star, \tau^\star)$ and rearranging the resulting inequality yields that

$$\begin{pmatrix} \tilde{\lambda}^{t+1} - \bar{\lambda}^t \\ \tilde{\tau}^{t+1} - \bar{\tau}^t \end{pmatrix}^\top \nabla\varphi(\bar{\lambda}^t, \bar{\tau}^t) + 2\theta_t \left( \sum_{k=1}^{m} \omega_k \left\| \begin{pmatrix} \tilde{\lambda}_k^{t+1} - \tilde{\lambda}_k^t \\ \tilde{\tau}_k^{t+1} - \tilde{\tau}_k^t \end{pmatrix} \right\|^2 \right)$$

$$\leq \begin{pmatrix} \lambda^\star - \bar{\lambda}^t \\ \tau^\star - \bar{\tau}^t \end{pmatrix}^\top \nabla\varphi(\bar{\lambda}^t, \bar{\tau}^t) + 2\theta_t \left( \sum_{k=1}^{m} \omega_k \left( \left\| \begin{pmatrix} \lambda_k^\star - \tilde{\lambda}_k^t \\ \tau_k^\star - \tilde{\tau}_k^t \end{pmatrix} \right\|^2 - \left\| \begin{pmatrix} \lambda_k^\star - \tilde{\lambda}_k^{t+1} \\ \tau_k^\star - \tilde{\tau}_k^{t+1} \end{pmatrix} \right\|^2 \right) \right).$$

Using the convexity of $\varphi$ again, we have

$$\begin{pmatrix} \lambda^\star - \bar{\lambda}^t \\ \tau^\star - \bar{\tau}^t \end{pmatrix}^\top \nabla\varphi(\bar{\lambda}^t, \bar{\tau}^t) \leq \varphi(\lambda^\star, \tau^\star) - \varphi(\bar{\lambda}^t, \bar{\tau}^t).$$

Putting these pieces together yields that

$$\text{I} \leq \varphi(\lambda^\star, \tau^\star) + 2\theta_t \left( \sum_{k=1}^{m} \omega_k \left( \left\| \begin{pmatrix} \lambda_k^\star - \tilde{\lambda}_k^t \\ \tau_k^\star - \tilde{\tau}_k^t \end{pmatrix} \right\|^2 - \left\| \begin{pmatrix} \lambda_k^\star - \tilde{\lambda}_k^{t+1} \\ \tau_k^\star - \tilde{\tau}_k^{t+1} \end{pmatrix} \right\|^2 \right) \right). \tag{23}$$

Plugging Eq. (22) and Eq. (23) into Eq. (21) yields that

$$\varphi(\widehat{\lambda}^{t+1}, \widehat{\tau}^{t+1}) \leq (1-\theta_t)\varphi(\check{\lambda}^t, \check{\tau}^t) + \theta_t\varphi(\lambda^\star, \tau^\star) + 2\theta_t^2 \left( \sum_{k=1}^{m} \omega_k \left( \left\| \begin{pmatrix} \lambda_k^\star - \tilde{\lambda}_k^t \\ \tau_k^\star - \tilde{\tau}_k^t \end{pmatrix} \right\|^2 - \left\| \begin{pmatrix} \lambda_k^\star - \tilde{\lambda}_k^{t+1} \\ \tau_k^\star - \tilde{\tau}_k^{t+1} \end{pmatrix} \right\|^2 \right) \right).$$

Since $(\check{\lambda}^{t+1}, \check{\tau}^{t+1})$ is obtained by an exact coordinate update from $(\lambda^t, \tau^t)$, we have $\varphi(\lambda^t, \tau^t) \geq \varphi(\check{\lambda}^{t+1}, \check{\tau}^{t+1})$. Using the similar argument, we have $\varphi(\acute{\lambda}^t, \acute{\tau}^t) \geq \varphi(\grave{\lambda}^t, \grave{\tau}^t) \geq \varphi(\lambda^t, \tau^t)$. By the definition of $(\acute{\lambda}^t, \acute{\tau}^t)$, we have $\varphi(\widehat{\lambda}^t, \widehat{\tau}^t) \geq \varphi(\acute{\lambda}^t, \acute{\tau}^t)$. Putting these pieces together yields the desired inequality.

## E.5 Proof of Theorem 4.2

First, let $\delta_t = \varphi(\check{\lambda}^t, \check{\tau}^t) - \varphi(\lambda^\star, \tau^\star)$, we show that

$$\delta_t \leq \frac{8n(R_\lambda^2 + R_\tau^2)}{(t+1)^2}. \tag{24}$$

Indeed, by Lemma E.1 and E.4, we have

$$\left(\frac{1 - \theta_{t+1}}{\theta_{t+1}^2}\right)\delta_{t+1} - \left(\frac{1 - \theta_t}{\theta_t^2}\right)\delta_t \leq 2\left(\sum_{k=1}^m \omega_k \left(\left\|\begin{pmatrix} \lambda_k^\star - \tilde{\lambda}_k^t \\ \tau_k^\star - \tilde{\tau}_k^t \end{pmatrix}\right\|^2 - \left\|\begin{pmatrix} \lambda_k^\star - \tilde{\lambda}_k^{t+1} \\ \tau_k^\star - \tilde{\tau}_k^{t+1} \end{pmatrix}\right\|^2\right)\right).$$

By unrolling the recurrence and using $\theta_0 = 1$ and $\tilde{\lambda}_0 = \tilde{\tau}_0 = \mathbf{0}_{mn}$, we have

$$\left(\frac{1 - \theta_t}{\theta_t^2}\right)\delta_t + 2\left(\sum_{k=1}^m \omega_k \left\|\begin{pmatrix} \lambda_k^\star - \tilde{\lambda}_k^t \\ \tau_k^\star - \tilde{\tau}_k^t \end{pmatrix}\right\|^2\right) \leq \left(\frac{1 - \theta_0}{\theta_0^2}\right)\delta_0 + 2\left(\sum_{k=1}^m \omega_k \left\|\begin{pmatrix} \lambda_k^\star - \tilde{\lambda}_k^0 \\ \tau_k^\star - \tilde{\tau}_k^0 \end{pmatrix}\right\|^2\right)$$

$$\leq 2\left(\sum_{k=1}^m \omega_k \left\|\begin{pmatrix} \lambda_k^\star \\ \tau_k^\star \end{pmatrix}\right\|^2\right) \stackrel{\text{Corollary } 2.4}{\leq} 2n(R_\lambda^2 + R_\tau^2).$$

For $t \geq 1$, Lemma E.1 implies that $\theta_{t-1}^{-2} = (1 - \theta_t)\theta_t^{-2}$. Therefore, we conclude that

$$\delta_t \leq 2\theta_{t-1}^2 n(R_\lambda^2 + R_\tau^2).$$

This together with the fact that $0 < \theta_{t-1} \leq 2/(t+1)$ yields the desired inequality.

Furthermore, we show that

$$\delta_t - \delta_{t+1} \geq \frac{E_t^2}{11}. \tag{25}$$

Indeed, by the definition of $\Delta_t$, we have

$$\delta_t - \delta_{t+1} = \varphi(\check{\lambda}^t, \check{\tau}^t) - \varphi(\check{\lambda}^{t+1}, \check{\tau}^{t+1}) \geq \varphi(\lambda^t, \tau^t) - \varphi(\check{\lambda}^{t+1}, \check{\tau}^{t+1}).$$

By the definition of $\varphi$, we have

$$\varphi(\lambda^t, \tau^t) - \varphi(\check{\lambda}^{t+1}, \check{\tau}^{t+1}) = \sum_{k=1}^m \omega_k(\log(\|B_k(\lambda_k^t, \tau_k^t)\|_1) - \log(\|B_k(\check{\lambda}_k^{t+1}, \check{\tau}_k^{t+1})\|_1)).$$

Since $r(B_k(\lambda_k^t, \tau_k^t)) = u^k \in \Delta_n$ for all $k \in [m]$, we have $\|B_k(\lambda_k^t, \tau_k^t)\|_1 = 1$. This together with the update formula of $(\check{\lambda}^{t+1}, \check{\tau}^{t+1})$ yields that

$$\varphi(\lambda^t, \tau^t) - \varphi(\check{\lambda}^{t+1}, \check{\tau}^{t+1}) = -\log\left(\mathbf{1}_n^\top e^{\sum_{k=1}^m \omega_k \log(c(B_k(\lambda_k^t, \tau_k^t)))}\right).$$

Recall that $\log(1 + x) \leq x$ for all $x \in \mathbb{R}$, we have

$$\varphi(\lambda^t, \tau^t) - \varphi(\check{\lambda}^{t+1}, \check{\tau}^{t+1}) \geq 1 - \mathbf{1}_n^\top e^{\sum_{k=1}^m \omega_k \log(c(B_k(\lambda_k^t, \tau_k^t)))}.$$

Since $r(B_k(\lambda_k^t, \tau_k^t)) = u^k \in \Delta_n$ for all $k \in [m]$, we have $\mathbf{1}_n^\top c(B_k(\lambda_k^t, \tau_k^t)) = 1$. In addition, $(\omega_1, \omega_2, \ldots, \omega_m) \in \Delta^m$. Thus, we have

$$\varphi(\lambda^t, \tau^t) - \varphi(\check{\lambda}^{t+1}, \check{\tau}^{t+1}) \geq \mathbf{1}_n^\top \left(\sum_{k=1}^m \omega_k c(B_k(\lambda_k^t, \tau_k^t)) - e^{\sum_{k=1}^m \omega_k \log(c(B_k(\lambda_k^t, \tau_k^t)))}\right).$$

Combining $c(B_k(\lambda_k^t, \tau_k^t)) \in \Delta^n$ with the arguments in Kroshnin et al. [32, Lemma 6] yields

$$\varphi(\lambda^t, \tau^t) - \varphi(\check{\lambda}^{t+1}, \check{\tau}^{t+1}) \geq \frac{1}{11}\sum_{k=1}^m \omega_k \|c(B_k(\lambda_k^t, \tau_k^t)) - \sum_{i=1}^m \omega_i c(B_i(\lambda_i^t, \tau_i^t))\|_1^2.$$

Using the Cauchy-Schwarz inequality together with $\sum_{k=1}^m \omega_k = 1$, we have

$$E_t^2 \leq \sum_{k=1}^m \omega_k \|c(B_k(\lambda_k^t, \tau_k^t)) - \sum_{i=1}^m \omega_i c(B_i(\lambda_i, \tau_i))\|_1^2.$$

Putting these pieces together yields the desired inequality.

Finally, we derive from Eq. (24) and (25) and the non-negativeness of $\delta_t$ that

$$\sum_{i=t}^{+\infty} E_i^2 \leq 11 \left( \sum_{i=t}^{+\infty} (\delta_i - \delta_{i+1}) \right) \leq 11\delta_t \leq \frac{88n(R_\lambda^2 + R_\tau^2)}{(t+1)^2}$$

Let $T > 0$ satisfy $E_T \leq \varepsilon$, we have $E_t > \varepsilon$ for all $t \in [T]$. Without loss of generality, we assume $T$ is even. Then the following statement holds true:

$$\varepsilon^2 \leq \frac{704n(R_\lambda^2 + R_\tau^2)}{T^3}.$$

Rearranging the above inequality yields the desired inequality.

### E.6 Proof of Theorem 4.3

Consider the iterate $(\widetilde{X}_1, \widetilde{X}_2, \ldots, \widetilde{X}_m)$ be generated by the FASTIBP algorithm (cf. Algorithm 1), the rounding scheme (cf. Kroshnin et al. [32, Algorithm 4]) returns the feasible solution $(\widehat{X}_1, \widehat{X}_2, \ldots, \widehat{X}_m)$ to the FS-WBP in Eq. (3) and $c(\widehat{X}_k)$ are the same for all $k \in [m]$.

To show that $\widehat{u} = \sum_{k=1}^m \omega_k c(\widehat{X}_k)$ is an $\varepsilon$-approximate barycenter (cf. Definition 2.2), it suffices to show that

$$\sum_{k=1}^m \omega_k \langle C_k, \widehat{X}_k \rangle \leq \sum_{k=1}^m \omega_k \langle C_k, X_k^\star \rangle + \varepsilon, \tag{26}$$

where $(X_1^\star, X_2^\star, \ldots, X_m^\star)$ is a set of optimal transportation plan between $m$ measures $\{u^k\}_{k \in [m]}$ and the barycenter of the FS-WBP.

First, we derive from the scheme of Kroshnin et al. [32, Algorithm 4] that the following inequality holds for all $k \in [m]$,

$$\|\widehat{X}_k - \widetilde{X}_k\|_1 \leq \|c(\widetilde{X}_k) - \sum_{i=1}^m \omega_i c(\widetilde{X}_i)\|_1.$$

This together with the Hölder's inequality implies that

$$\sum_{k=1}^m \omega_k \langle C_k, \widehat{X}_k - \widetilde{X}_k \rangle \leq \left( \max_{1 \leq k \leq m} \|C_k\|_\infty \right) \left( \sum_{k=1}^m \omega_k \|c(\widetilde{X}_k) - \sum_{i=1}^m \omega_i c(\widetilde{X}_i)\|_1 \right). \tag{27}$$

Furthermore, we have

$$\sum_{k=1}^m \omega_k \langle C_k, \widetilde{X}_k - X_k^\star \rangle = \sum_{k=1}^m \omega_k (\langle C_k, \widetilde{X}_k \rangle - \eta H(\widetilde{X}_k)) - \sum_{k=1}^m \omega_k (\langle C_k, X_k^\star \rangle - \eta H(X_k^\star))$$

$$+ \sum_{k=1}^m \omega_k \eta H(\widetilde{X}_k) - \sum_{k=1}^m \omega_k \eta H(X_k^\star).$$

Since $0 \leq H(X) \leq 2\log(n)$ for any $X \in \mathbb{R}_+^{n \times n}$ satisfying that $\|X\|_1 = 1$ [17] and $\sum_{k=1}^m \omega_k = 1$, we have

$$\sum_{k=1}^m \omega_k \langle C_k, \widetilde{X}_k - X_k^\star \rangle \leq 2\eta \log(n) + \sum_{k=1}^m \omega_k (\langle C_k, \widetilde{X}_k \rangle - \eta H(\widetilde{X}_k)) - \sum_{k=1}^m \omega_k (\langle C_k, X_k^\star \rangle - \eta H(X_k^\star)).$$

Let $(X_1^\eta, X_2^\eta, \ldots, X_m^\eta)$ be a set of optimal transportation plans to the entropic regularized FS-WBP in Eq. (4), we have

$$\sum_{k=1}^m \omega_k (\langle C_k, X_k^\eta \rangle - \eta H(X_k^\eta)) \leq \sum_{k=1}^m \omega_k (\langle C_k, X_k^\star \rangle - \eta H(X_k^\star)).$$

By the optimality of $(X_1^\eta, X_2^\eta, \ldots, X_m^\eta)$, we have

$$\sum_{k=1}^m \omega_k (\langle C_k, X_k^\eta \rangle - \eta H(X_k^\eta)) = -\eta \left( \min_{\lambda \in \mathbb{R}^{mn}, \tau \in \mathcal{P}} \varphi(\lambda, \tau) \right) \geq -\eta \varphi(\lambda^t, \tau^t).$$

Since $(\widetilde{X}_1, \widetilde{X}_2, \ldots, \widetilde{X}_m)$ is generated by the FASTIBP algorithm, we have $\widetilde{X}_k = B_k(\lambda_k^t, \tau_k^t)$ for all $k \in [m]$ where $(\lambda^t, \tau^t)$ are the dual iterates. Then

$$
\begin{aligned}
\sum_{k=1}^m \omega_k(\langle C_k, \widetilde{X}_k \rangle - \eta H(\widetilde{X}_k)) &= \sum_{k=1}^m \omega_k(\langle C_k, B_k(\lambda_k^t, \tau_k^t) \rangle - \eta H(B_k(\lambda_k^t, \tau_k^t))) \\
&= -\eta \left( \sum_{k=1}^m \omega_k(\mathbf{1}_n^\top B_k(\lambda_k^t, \tau_k^t) \mathbf{1}_n - (\lambda_k^t)^\top u^k) \right) + \eta \sum_{k=1}^m \omega_k(\tau_k^t)^\top c(B_k(\lambda_k^t, \tau_k^t)) \\
&= -\eta \varphi(\lambda^t, \tau^t) + \eta \left( \sum_{k=1}^m \omega_k(\tau_k^t)^\top \left( c(B_k(\lambda_k^t, \tau_k^t)) - \sum_{i=1}^m \omega_i c(B_i(\lambda_i^t, \tau_i^t)) \right) \right).
\end{aligned}
$$

Putting these pieces together yields that

$$
\sum_{k=1}^m \omega_k \langle C_k, \widetilde{X}_k - X_k^\star \rangle \leq 2\eta \log(n) + \eta \left( \sum_{k=1}^m \omega_k(\tau_k^t)^\top \left( c(B_k(\lambda_k^t, \tau_k^t)) - \sum_{i=1}^m \omega_i c(B_i(\lambda_i^t, \tau_i^t)) \right) \right).
$$

Since $\mathbf{1}_n^\top c(B_k(\lambda_k^t, \tau_k^t)) = 1$ for all $k \in [m]$, we have

$$
\begin{aligned}
&\left( \sum_{k=1}^m \omega_k(\tau_k^t)^\top \left( c(B_k(\lambda_k^t, \tau_k^t)) - \sum_{i=1}^m \omega_i c(B_i(\lambda_i^t, \tau_i^t)) \right) \right) \\
&= \left( \sum_{k=1}^m \omega_k \left( \tau_k^t - \frac{\max_{1 \leq i \leq n}(\tau_k^t)_i + \min_{1 \leq i \leq n}(\tau_k^t)_i}{2} \mathbf{1}_n \right)^\top \left( c(B_k(\lambda_k^t, \tau_k^t)) - \sum_{i=1}^m \omega_i c(B_i(\lambda_i^t, \tau_i^t)) \right) \right) \\
&\leq \left\| \tau_k^t - \frac{\max_{1 \leq i \leq n}(\tau_k^t)_i + \min_{1 \leq i \leq n}(\tau_k^t)_i}{2} \mathbf{1}_n \right\|_\infty \left( \sum_{k=1}^m \omega_k \| c(\widetilde{X}_k) - \sum_{i=1}^m \omega_i c(\widetilde{X}_i) \|_1 \right).
\end{aligned}
$$

Using Lemma E.3, we have

$$
\left\| \tau_k^t - \frac{\max_{1 \leq i \leq n}(\tau_k^t)_i + \min_{1 \leq i \leq n}(\tau_k^t)_i}{2} \mathbf{1}_n \right\|_\infty \leq \frac{R_\tau}{2}.
$$

Putting these pieces together yields that

$$
\sum_{k=1}^m \omega_k \langle C_k, \widetilde{X}_k - X_k^\star \rangle \leq 2\eta \log(n) + \frac{\eta R_\tau}{2} \left( \sum_{k=1}^m \omega_k \| c(\widetilde{X}_k) - \sum_{i=1}^m \omega_i c(\widetilde{X}_i) \|_1 \right). \tag{28}
$$

Recall that $E_t = \sum_{k=1}^m \omega_k \| c(\widetilde{X}_k) - \sum_{i=1}^m \omega_i c(\widetilde{X}_i) \|_1$ and $R_\tau = 4\eta^{-1}(\max_{1 \leq k \leq m} \|C_k\|_\infty)$. Then Eq. (27) and Eq. (28) together imply that

$$
\sum_{k=1}^m \omega_k \langle C_k, \widehat{X}_k - X_k^\star \rangle \leq 2\eta \log(n) + 3 \left( \max_{1 \leq k \leq m} \|C_k\|_\infty \right) E_t.
$$

This together with $E_t \leq \bar{\varepsilon}/2$ and the choice of $\eta$ and $\bar{\varepsilon}$ implies Eq. (26) as desired.

**Complexity bound estimation.** We first bound the number of iterations required by the FASTIBP algorithm (cf. Algorithm 1) to reach $E_t \leq \bar{\varepsilon}/2$. Indeed, Theorem 4.2 implies that

$$
t \leq 1 + 20 \left( \frac{n(R_\lambda^2 + R_\tau^2)}{\bar{\varepsilon}^2} \right)^{1/3} \leq 20 \sqrt[3]{n} \left( \frac{R_\lambda + R_\tau}{\bar{\varepsilon}} \right)^{2/3}.
$$

For the simplicity, we let $\bar{C} = \max_{1 \leq k \leq m} \|C_k\|_\infty$. Using the definition of $R_\lambda$ and $R_\tau$ in Lemma 2.2, the construction of $\{\tilde{u}^k\}_{k \in [m]}$ and the choice of $\eta$ and $\bar{\varepsilon}$, we have

$$
\begin{aligned}
t &\leq 1 + 20 \sqrt[3]{n} \left( \frac{4\bar{C}}{\varepsilon} \left( \frac{36 \log(n)\bar{C}}{\varepsilon} + \log(n) - \log \left( \frac{16n\bar{C}}{\varepsilon} \right) \right) \right)^{2/3} \\
&= O \left( \sqrt[3]{n} \left( \frac{\bar{C}\sqrt{\log(n)}}{\varepsilon} \right)^{4/3} \right).
\end{aligned}
$$

Recall that each iteration of the FASTIBP algorithm requires $\mathcal{O}(mn^2)$ arithmetic operations, the total arithmetic operations required by the FASTIBP algorithm as the subroutine in Algorithm 2 is bounded by

$$O\left(mn^{7/3}\left(\frac{\bar{C}\sqrt{\log(n)}}{\varepsilon}\right)^{4/3}\right).$$

Computing a collection of vectors $\{\tilde{u}^k\}_{k\in[m]}$ needs $\mathcal{O}(mn)$ arithmetic operations while the rounding scheme in Kroshnin et al. [32, Algorithm 4] requires $\mathcal{O}(mn^2)$ arithmetic operations. Putting these pieces together yields that the desired complexity bound of Algorithm 2.

## F   Implementation Details

All the experiments are conducted in MATLAB R2020a on a workstation with an Intel Core i5-9400F (6 cores and 6 threads) and 32GB memory, equipped with Ubuntu 18.04. For a fair comparison, we do not implement convolutional technique [50] and its stabilized version [49, Section 4.1.2], which can be used to substantially improve IBP with small $\eta$.

For the FASTIBP algorithm, the regularization parameter $\eta$ is chosen from $\{0.01, 0.001\}$ in our experiments. We follow Benamou et al. [4, Remark 3] to implement the algorithm and terminate it when

$$\frac{\sum_{k=1}^m \omega_k \|c(B_k(\lambda_k^t, \tau_k^t)) - \sum_{i=1}^m \omega_i c(B_i(\lambda_i^t, \tau_i^t))\|}{1 + \sum_{k=1}^m \omega_k \|c(B_k(\lambda_k^t, \tau_k^t))\| + \|\sum_{i=1}^m \omega_i c(B_i(\lambda_i^t, \tau_i^t))\|} \leq \mathsf{Tol}_{\mathsf{fibp}},$$

$$\frac{\sum_{k=1}^m \omega_k \|r(B_k(\lambda_k^t, \tau_k^t)) - u^k\|}{1 + \sum_{k=1}^m \omega_k \|r(B_k(\lambda_k^t, \tau_k^t))\| + \sum_{k=1}^m \omega_k \|u^k\|} \leq \mathsf{Tol}_{\mathsf{fibp}},$$

$$\frac{\|\sum_{i=1}^m \omega_i c(B_i(\lambda_i^t, \tau_i^t)) - \sum_{i=1}^m \omega_i c(B_i(\lambda_i^{t-1}, \tau_i^{t-1}))\|}{1 + \|\sum_{i=1}^m \omega_i c(B_i(\lambda_i^t, \tau_i^t))\| + \|\sum_{i=1}^m \omega_i c(B_i(\lambda_i^{t-1}, \tau_i^{t-1}))\|} \leq \mathsf{Tol}_{\mathsf{fibp}},$$

$$\frac{\sum_{k=1}^m \omega_k \|B_k(\lambda_k^t, \tau_k^t) - B_k(\lambda_k^{t-1}, \tau_k^{t-1})\|_F}{1 + \sum_{k=1}^m \omega_k \|B_k(\lambda_k^t, \tau_k^t)\|_F + \sum_{k=1}^m \omega_k \|B_k(\lambda_k^{t-1}, \tau_k^{t-1})\|_F} \leq \mathsf{Tol}_{\mathsf{fibp}},$$

$$\frac{\sum_{k=1}^m \omega_k \|\lambda_k^t - \lambda_k^{t-1}\|}{1 + \sum_{k=1}^m \omega_k \|\lambda_k^t\| + \sum_{k=1}^m \omega_k \|\lambda_k^{t-1}\|} \leq \mathsf{Tol}_{\mathsf{fibp}},$$

$$\frac{\sum_{k=1}^m \omega_k \|\tau_k^t - \tau_k^{t-1}\|}{1 + \sum_{k=1}^m \omega_k \|\tau_k^t\| + \sum_{k=1}^m \omega_k \|\tau_k^{t-1}\|} \leq \mathsf{Tol}_{\mathsf{fibp}}.$$

These inequalities guarantee that (i) the infeasibility violations for marginal constraints, (ii) the iterative gap between approximate barycenters, and (iii) the iterative gap between dual variables are relatively small. Computing all the above residuals is expensive. Thus, in our implementations, we only compute them and check the termination criteria at every 20 iterations when $\eta = 0.01$ and every 200 iteration when $\eta = 0.001$. We set $\mathsf{Tol}_{\mathsf{fibp}} = 10^{-6}$ and $\mathsf{MaxIter}_{\mathsf{fibp}} = 10000$ on synthetic data and $\mathsf{Tol}_{\mathsf{fibp}} = 10^{-10}$ on MNIST images.

For IBP and BADMM, we use the Matlab code[2] implemented by Ye et al. [57] and terminate them with the refined stopping criterion provided by [56]. The regularization parameter $\eta$ for the IBP algorithm is still chosen from $\{0.01, 0.001\}$. For synthetic data, we set $\mathsf{Tol}_{\mathsf{badmm}} = 10^{-5}$ and $\mathsf{Tol}_{\mathsf{ibp}} = 10^{-6}$ with $\mathsf{MaxIter}_{\mathsf{badmm}} = 5000$ and $\mathsf{MaxIter}_{\mathsf{ibp}} = 10000$. For MNIST images, we set $\mathsf{Tol}_{\mathsf{ibp}} = 10^{-10}$.

For the linear programming algorithm, we apply Gurobi 9.0.2 (Gurobi Optimization, 2019) (with an academic license) to solve the FS-WBP in Eq. (3). Since Gurobi can provide high quality solutions when the problem of medium size, we use the solution obtained by Gurobi as a benchmark to evaluate the qualities of solution obtained by different algorithms on synthetic data. In our experiments, we force Gurobi to *only run the dual simplex algorithm* and use other parameters in the default settings.

For the evaluation metrics, "**normalized obj**" stands for the normalized objective value which is defined by

$$\text{normalized obj} := \frac{|\sum_{k=1}^{m} \omega_k \langle C_k, \widehat{X}_k \rangle - \sum_{k=1}^{m} \omega_k \langle C_k, X_k^{\mathsf{g}} \rangle|}{|\sum_{k=1}^{m} \omega_k \langle C_k, X_k^{\mathsf{g}} \rangle|},$$

where $(\widehat{X}_1, \ldots, \widehat{X}_m)$ is the solution obtained by each algorithm and $(X_1^{\mathsf{g}}, \ldots, X_m^{\mathsf{g}})$ denotes the solution obtained by Gurobi. "**feasibility**" denotes the the deviation of the terminating solution from the feasible set[3]; see Yang et al. [56, Section 5.1]. "**iteration**" denotes the number of iterations. "**time (in seconds)**" denotes the computational time.

## G    Experimental Setup and Additional Results

**Generating synthetic data.**    We generate a set of discrete probability distributions $\{\mu_k\}_{k=1}^{m}$ with $\mu_k = \{(u_i^k, \mathbf{x}_i) \in \mathbb{R}_+ \times \mathbb{R}^d \mid i \in [n]\}$ and $\sum_{i=1}^{n} u_i^k = 1$. The fixed-support Wasserstein barycenter $\widehat{\mu} = \{(\widehat{u}_i, \mathbf{x}_i) \in \mathbb{R}_+ \times \mathbb{R}^d \mid i \in [n]\}$ where $(\mathbf{x}_1, \mathbf{x}_2, \ldots, \mathbf{x}_n)$ are known. In our experiment, we set $d = 3$ and choose different values of $(m, n)$. Then, given each tuple $(m, n)$, we randomly generate a trial as follows.

First, we generate the support points $(\mathbf{x}_1^k, \mathbf{x}_2^k, \ldots, \mathbf{x}_n^k)$ whose entries are drawn from a Gaussian mixture distribution via the Matlab commands provided by Yang et al. [56]:

```
gm_num = 5; gm_mean = [-20; -10; 0; 10; 20];
sigma = zeros(1, 1, gm_num); sigma(1, 1, :) = 5*ones(gm_num, 1);
gm_weights = rand(gm_num, 1); gm_weights =
gm_weights/sum(gm_weights);
distrib = gmdistribution(gm_mean, sigma, gm_weights);
```

For each $k \in [m]$, we generate the weight vector $(u_1^k, u_2^k, \ldots, u_n^k)$ whose entries are drawn from the uniform distribution on the interval $(0, 1)$, and normalize it such that $\sum_{i=1}^{n} u_i^k = 1$. After generating all $\{\mu_k\}_{k=1}^{m}$, we use the k-means[4] method to choose $n$ points from $\{\mathbf{x}_i^k \mid i \in [n], k \in [m]\}$ to be the support points of the barycenter. Finally, we generate the weight vector $(\omega_1, \omega_2, \ldots, \omega_m)$ whose entries are drawn from the uniform distribution on the interval $(0, 1)$, and normalize by $\sum_{k=1}^{m} \omega_k = 1$.

**Processing MNIST images.**    To better visualize the quality of approximate barycenters obtained by each algorithm, we follow Cuturi and Doucet [19] on the MNIST[5] dataset [35]. We randomly select 50 images for each digit (1∼9) and resize each image to $\zeta$ times of its original size of $28 \times 28$, where $\zeta$ is drawn uniformly at random from $[0.5, 2]$. We randomly put each resized image in a larger $56 \times 56$ blank image and normalize the resulting image so that all pixel values add up to 1. Each image can be viewed as a discrete distribution supported on grids. Additionally, we set the weight vector $(\omega_1, \omega_2, \ldots, \omega_m)$ such that $\omega_k = 1/m$ for all $k \in [m]$. The size of barycenter is set to $56 \times 56$.

**Additional results on synthetic data.**    To facilitate the readers, we present the averaged results from 10 independent trials with FASTIBP, IBP, BADMM algorithms, and Gurobi in Table 2. Note that we implement *the rounding scheme* after each algorithm (except Gurobi) so the terms in **"feasibility"** are zero *up to numerical errors* for most of medium-size problems.

Table 2: Numerical results on synthetic data where each distribution has different dense weights but same support size. The support points of the barycenter is fixed.

| m | n | g | b | i1 | i2 | f1 | f2 |
|---|---|---|---|---|---|---|---|
| | | | | **normalized obj** | | | |
| 20 | 50 | - | 5.9e-01±1.2e-01 | 2.0e-01±8.0e-02 | 2.1e-01±1.1e-01 | 5.7e-02±1.1e-02 | 1.7e-03±9.7e-04 |
| 20 | 100 | - | 6.7e-01±8.2e-02 | 3.2e-01±5.5e-02 | 3.6e-01±9.5e-02 | 6.7e-02±8.0e-03 | 2.1e-03±8.2e-04 |
| 20 | 200 | - | 7.8e-01±7.4e-02 | 4.8e-01±5.9e-02 | 6.0e-01±7.6e-02 | 6.3e-02±4.7e-03 | 2.9e-03±3.8e-04 |
| 50 | 50 | - | 4.5e-01±4.3e-02 | 1.7e-01±3.7e-02 | 1.6e-01±4.9e-02 | 6.8e-02±1.0e-02 | 2.2e-03±8.3e-04 |
| 50 | 100 | - | 6.4e-01±1.0e-01 | 3.7e-01±6.8e-02 | 4.3e-01±6.4e-02 | 7.6e-02±8.3e-03 | 3.0e-03±6.1e-04 |
| 50 | 200 | - | 8.2e-01±7.8e-02 | 5.9e-01±5.7e-02 | 6.7e-01±8.5e-02 | 6.2e-02±6.9e-03 | 4.0e-03±6.4e-04 |
| 100 | 50 | - | 3.1e-01±3.0e-02 | 1.1e-01±2.9e-02 | 7.2e-02±2.8e-02 | 6.7e-02±1.4e-02 | 3.9e-03±2.3e-03 |
| 100 | 100 | - | 6.1e-01±9.3e-02 | 3.8e-01±6.0e-02 | 4.6e-01±7.2e-02 | 7.7e-02±6.0e-03 | 3.6e-03±7.0e-04 |
| 100 | 200 | - | 8.3e-01±5.0e-02 | 6.1e-01±4.0e-02 | 7.5e-01±4.2e-02 | 5.6e-02±4.7e-03 | 4.3e-03±6.9e-04 |
| 200 | 50 | - | 2.8e-01±4.2e-02 | 1.1e-01±3.9e-02 | 6.0e-02±3.8e-02 | 6.9e-02±1.4e-02 | 3.2e-03±1.8e-03 |
| 200 | 100 | - | 4.4e-01±4.6e-02 | 2.8e-01±3.0e-02 | 3.7e-01±5.8e-02 | 7.9e-02±3.1e-03 | 3.7e-03±3.3e-04 |
| 200 | 200 | - | 8.0e-01±8.7e-02 | 6.0e-01±6.8e-02 | 7.2e-01±4.6e-02 | 5.7e-02±4.5e-03 | 5.2e-03±4.7e-04 |
| | | | | **feasibility** | | | |
| 20 | 50 | 4.9e-07±0.0e+00 | 1.9e-07±0.0e+00 | 9.0e-07±0.0e+00 | 9.6e-07±0.0e+00 | 1.8e-14±0.0e+00 | 3.6e-09±0.0e+00 |
| 20 | 100 | 0.0e+00±0.0e+00 | 0.0e+00±0.0e+00 | 0.0e+00±0.0e+00 | 0.0e+00±0.0e+00 | 0.0e+00±0.0e+00 | 0.0e+00±0.0e+00 |
| 20 | 200 | 0.0e+00±0.0e+00 | 0.0e+00±0.0e+00 | 0.0e+00±0.0e+00 | 0.0e+00±0.0e+00 | 0.0e+00±0.0e+00 | 0.0e+00±0.0e+00 |
| 50 | 50 | 0.0e+00±0.0e+00 | 0.0e+00±0.0e+00 | 0.0e+00±0.0e+00 | 0.0e+00±0.0e+00 | 0.0e+00±0.0e+00 | 0.0e+00±0.0e+00 |
| 50 | 100 | 0.0e+00±0.0e+00 | 0.0e+00±0.0e+00 | 0.0e+00±0.0e+00 | 0.0e+00±0.0e+00 | 0.0e+00±0.0e+00 | 0.0e+00±0.0e+00 |
| 50 | 200 | 0.0e+00±0.0e+00 | 0.0e+00±0.0e+00 | 0.0e+00±0.0e+00 | 0.0e+00±0.0e+00 | 0.0e+00±0.0e+00 | 0.0e+00±0.0e+00 |
| 100 | 50 | 0.0e+00±0.0e+00 | 0.0e+00±0.0e+00 | 0.0e+00±0.0e+00 | 0.0e+00±0.0e+00 | 0.0e+00±0.0e+00 | 0.0e+00±0.0e+00 |
| 100 | 100 | 0.0e+00±0.0e+00 | 0.0e+00±0.0e+00 | 0.0e+00±0.0e+00 | 0.0e+00±0.0e+00 | 0.0e+00±0.0e+00 | 0.0e+00±0.0e+00 |
| 100 | 200 | 0.0e+00±0.0e+00 | 0.0e+00±0.0e+00 | 0.0e+00±0.0e+00 | 0.0e+00±0.0e+00 | 0.0e+00±0.0e+00 | 0.0e+00±0.0e+00 |
| 200 | 50 | 0.0e+00±0.0e+00 | 0.0e+00±0.0e+00 | 0.0e+00±0.0e+00 | 0.0e+00±0.0e+00 | 0.0e+00±0.0e+00 | 0.0e+00±0.0e+00 |
| 200 | 100 | 0.0e+00±0.0e+00 | 0.0e+00±0.0e+00 | 0.0e+00±0.0e+00 | 0.0e+00±0.0e+00 | 0.0e+00±0.0e+00 | 0.0e+00±0.0e+00 |
| 200 | 200 | 2.7e-07±1.9e-07 | 2.2e-07±2.3e-08 | 6.2e-07±1.5e-07 | 9.2e-07±3.9e-08 | 7.5e-14±1.9e-13 | 1.0e-07±1.6e-07 |
| | | | | **iteration** | | | |
| 20 | 50 | 3115±532 | 5000±0 | 440±158 | 7140±3912 | 200±0 | 2760±1560 |
| 20 | 100 | 6415±999 | 3400±94 | 400±0 | 8660±5464 | 200±0 | 2860±1678 |
| 20 | 200 | 12430±2339 | 3580±175 | 400±0 | 5520±2368 | 200±0 | 1240±497 |
| 50 | 50 | 10139±1192 | 5000±0 | 360±227 | 7320±6351 | 200±0 | 3560±1786 |
| 50 | 100 | 21697±4377 | 3480±103 | 580±63 | 9920±5351 | 200±0 | 1540±1116 |
| 50 | 200 | 39564±6916 | 3740±97 | 580±63 | 6800±2512 | 220±63 | 1240±833 |
| 100 | 50 | 26729±2731 | 4580±887 | 380±274 | 5180±1901 | 240±84 | 5940±2406 |
| 100 | 100 | 52799±9610 | 3560±84 | 700±287 | 10020±3400 | 200±0 | 1360±815 |
| 100 | 200 | 97357±10615 | 3780±114 | 920±103 | 9720±3737 | 200±0 | 440±280 |
| 200 | 50 | 55841±6359 | 4280±1038 | 300±141 | 11140±4724 | 200±0 | 8000±3749 |
| 200 | 100 | 149230±18051 | 3600±0 | 980±537 | 12840±3650 | 200±0 | 1880±634 |
| 200 | 200 | 258059±62104 | 3800±94 | 1440±158 | 12160±2609 | 200±0 | 340±165 |
| | | | | **time (in seconds)** | | | |
| 20 | 50 | 3.6e-01±1.1e-01 | 9.4e+00±3.6e+00 | 2.5e-01±1.1e-01 | 4.1e+00±2.2e+00 | 3.1e-01±5.7e-02 | 5.2e+00±2.4e+00 |
| 20 | 100 | 1.9e+00±1.2e+00 | 2.4e+01±3.9e+00 | 1.2e+00±6.2e-01 | 2.5e+01±1.5e+01 | 1.6e+00±9.0e-01 | 2.1e+01±1.3e+01 |
| 20 | 200 | 6.2e+00±1.7e+00 | 1.2e+02±5.1e+00 | 4.0e+00±5.6e-01 | 5.9e+01±2.7e+01 | 5.9e+00±7.5e-01 | 3.9e+01±1.5e+01 |
| 50 | 50 | 3.1e+00±1.3e+00 | 2.3e+01±4.8e+00 | 5.4e-01±3.8e-01 | 1.2e+01±1.1e+01 | 1.1e+00±6.0e-01 | 1.8e+01±9.2e+00 |
| 50 | 100 | 1.1e+01±2.1e+00 | 6.9e+01±5.0e+00 | 3.7e+00±7.1e-01 | 7.2e+01±4.1e+01 | 3.5e+00±5.3e-01 | 3.0e+01±2.2e+01 |
| 50 | 200 | 2.7e+01±5.8e+00 | 3.2e+02±1.4e+01 | 1.7e+01±5.2e+00 | 2.0e+02±7.5e+01 | 1.5e+01±4.5e+00 | 8.8e+01±5.5e+01 |
| 100 | 50 | 1.3e+01±4.3e+00 | 4.3e+01±7.8e+00 | 1.1e+00±9.0e-01 | 1.7e+01±7.0e+00 | 2.7e+00±1.2e+00 | 6.1e+01±2.7e+01 |
| 100 | 100 | 3.6e+01±1.1e+01 | 1.4e+02±3.9e+00 | 7.9e+00±3.5e+00 | 1.4e+02±4.9e+01 | 7.2e+00±1.0e+00 | 5.2e+01±3.0e+01 |
| 100 | 200 | 1.0e+02±2.1e+01 | 6.6e+02±2.7e+01 | 5.1e+01±6.0e+00 | 5.7e+02±2.3e+02 | 2.7e+01±3.8e+00 | 6.6e+01±4.1e+01 |
| 200 | 50 | 5.4e+01±1.2e+01 | 9.3e+01±2.5e+01 | 2.0e+00±8.9e-01 | 9.0e+01±4.2e+01 | 4.8e+00±2.9e+00 | 1.8e+02±1.0e+02 |
| 200 | 100 | 2.8e+02±6.7e+01 | 3.2e+02±2.7e+01 | 3.0e+01±1.9e+01 | 4.0e+02±1.3e+02 | 1.5e+01±4.5e+00 | 1.5e+02±5.1e+01 |
| 200 | 200 | 4.9e+02±2.0e+02 | 1.9e+03±9.5e+01 | 2.8e+02±3.2e+01 | 2.5e+03±5.6e+02 | 1.1e+02±8.1e+00 | 1.9e+02±9.5e+01 |

## Footnotes

[2]Available in https://github.com/bobye/WBC_Matlab

[3]Since we do not put the iterative gap between dual variables in "**feasibility**" and the FS-WBP is relatively easier than general WBP, our results for BADMM and IBP are consistently smaller than that presented by [57, 56, 26].

[4]In our experiments, we call the Matlab function kmeans, which is built in machine learning toolbox.

[5]Available in http://yann.lecun.com/exdb/mnist/