[Reviews · NeurIPS 2020]

Review 1

Summary and Contributions: The authors analyze the problem of computing wasserstein barycenters with fixed discrete support. They theoretically demonstrate that while computing the barycenter of m=2 histograms can be cast as a minimum cost flow problem (MCF), when m>=3, it cannot be cast as such, rendering off-the-shelf MCF solvers inefficient. The authors then introduce an algorithm that utilizes entropic regularization to yield an approximate solution to the wasserstein barycenter problem in state of the art complexity.

Strengths: The main strengths of this paper are the theoretical analysis and the fact that the authors introduce a considerably faster algorithm than is previously been presented. The demonstration that the FS-WBP is not an MCF is pedagogical and the combinatorial proof techniques may be of use in other OT problems. The major result of the paper is the algorithmic complexity improvement and that is quantified well in the numerical experiments.

Weaknesses: The main weaknesses are the lack of quantitative and qualitative evaluations. As the authors have pointed out, there are many applications of WSB in image and shape analysis and computer vision in general. The results on MNIST while there are many other applications, leaves something to be desired.

Correctness: To best of my ability, the proofs are sound and correct.

Clarity: Some of the notation in the main text are not self contained i.e. r( ) and c( ) had to be inferred from the supplementary.

Relation to Prior Work: While there have been several attempts to introduce fast algorithms to compute wasserstein barycenters, the authors compare against the best two in terms of complexity in n and \epsilon (IBP and accelarated IBP, respectively) and show that they outperform both.

Reproducibility: Yes

Additional Feedback:


Review 2

Summary and Contributions: This paper studies the FS- WBP, in particular whether it is minimum cost flow (MCF) problem (which is not for m >= 3, n>=3), and proposes a new fast IBP algorithm while studying its complexity and convergence accuracy. The first result is relevant as it justifies the entropic regularized barycenter problem as a computationally efficient algorithm. The algorithm proposal complements the analysis with improved accuracy and convergence results w.r.t. IBP.

Strengths: The discussion about totally unimodular matrices and MCF equivalence is interesting and justifies either looking for other LP efficient reformulations, or approximate methods, which is of big interest to solve the problem. I find the new algorithm proposal also of interest, although more clarity on the derivation of the algorithm and comparisons could improve the paper.

Weaknesses: I find the numerical experiments a bit lacking, even though the authors claim their experiments are extensive (which is a subjective expression and open to criticism). For example, I miss comparisons with [31] and [55], which should be very competitive. Even if [55] is not an entropic regularized formulation, it may still be very competitive. Comparing with [55] may justify why an entropic regularized formulation is needed. Also, why not compare with prox-IBP from [31]? Simulations with Gurobi are nice to see, but they are almost unnecessary. It is known that general solvers scale badly with dimensions and that's why specialized algorithms are useful. Regarding the MNIST visualization, I did not understand why the authors claimed FastIBP provide a "smoother" solution, while I think it is sharper (and sharper should be better, more visually appealing and closer to the unregularized barycenter?). I was surprised in Algorithm 2 by steps 1 and 3, and I would like such steps be better explained in text. Notation should definitely not be in the appendix, which I did not find until specifically looking for something on the appendix.

Correctness: The claims and methods are probably correct, although I did not verify the proofs carefully. The empirical methodology is correct, although I think more comparisons would improve the paper quality.

Clarity: The paper is well written. A few typos: -page 3 line 114: "progress has been, (...)"

Relation to Prior Work: Prior work seems to be thorough and is clearly discussed.

Reproducibility: Yes

Additional Feedback: After rebuttal: Thanks for your response and addressing my concerns. I have increased my evaluation to accept. Please, add the new comparisons you mention in your response to the final version.


Review 3

Summary and Contributions: The paper proves the Wasserstein barycenter problem is not a minimum cost flow problem, and develops a faster regularized algorithm than the currently popular Sinkhorn method for computing the fixed support barycenter.

Strengths: Well supported theoretical claims that form a significant contribution to the theory of Wasserstein barycenters. The authors additionally propose an improved algorithm for computing regularized barycenters that is both faster and more efficient than the Sinkhorn method.

Weaknesses: This paper does not benefit from the short form and limited number of pages of NeurIPS and would be better suited for a journal. The two main contributions of the paper are Theorems 3.5 and 4.3 whose proofs take several pages of the supplemental. The numerical verification is quite weak (again, likely due to space limitations) as only two setups are considered. I do not think these issues are significant enough to recommend rejection, but I would like to see either more intuition, or more empirical verification in a revision.

Correctness: To my understanding, every statement in the paper is correct. The proofs are detailed and easy to follow.

Clarity: The paper is clearly written, but very terse. Much of the interesting content is hidden in the supplemental, and the paper is rife with abbreviations and rushed definitions and statements.

Relation to Prior Work: All relevant prior work is acknowledged.

Reproducibility: Yes

Additional Feedback: After rebuttal: The authors address my concerns on empirical testing. My score will not change, as I already wish to see the paper accepted.


Review 4

Summary and Contributions: This paper considers approximation in the fixed support Wasserstein barycenter problem. For this problem, one wishes to find the barycenter of a set of discrete probability measures. Instead of letting the support of the discrete barycenter vary, instead one fixes a support beforehand and solves an optimization problem over an n-dimensional simplex. While the OT problem can be cast as a min-cost-flow problem, the authors resolve an open question and show that the fixed support Wasserstein barycenter problem in general is not. Therefore, other algorithmic solutions that are not optimal for the MCF problem must be pursued. The authors propose to solve the linear program efficiently using entropic regularization as well as an accelerated iterative Bregman projection (FastIBP) scheme. They justify this algorithm theoretically by showing that it achieves comparable approximation accuracy with some state-of-the-art algorithms. They also give some experiments that show the tradeoff achieved by the method: fast speed with good accuracy. Finally, an experiment on the MNIST data shows that this algorithm converges very quickly for small regularization, which allows them to achieve sharp barycenters on each digit.

Strengths: - The work resolves an important open question and shows that the fixed support Wasserstein barycenter problem is not a minimum cost flow. This is nontrivial, and the authors give a sketch of how they manage to prove this using some classical combinatorial theorems as well as carefully writing out the constraints in the fixed support Wasserstein barycenter problem. They give a specific counterexample in the case of n=3 (number of points) and m=3 (number of measures). - Since other algorithms are needed, the authors propose a new algorithm based on entropic regularization that achieves a novel theoretical complexity bound -- its dependence on the approximation parameter epsilon is better than the plain iterative Bregman projection (IBP) algorithm. - Experiments show that the proposed FastIBP method is faster than existing methods while achieving good approximation accuracy. In the regime tested, the FastIBP algorithm seems to perform better than other standards, such as IBP and BADMM. -The experiments on MNIST show that the algorithm gives sharper barycenters than the non-accelerated iterative Bergman projection algorithm in a fixed amount of time.

Weaknesses: - The setting is restrictive, as it requires a setting where the barycenter support must be pre-specified. In general, the ideas for solving the fixed support problem do not seem to generalize to the general Wasserstein barycenter problem. - The algorithm they propose does not achieve the best complexity bound in all regimes. There seems to be a tradeoff between approximation factor and size of the empirical distributions, since it is obviously faster than IBP (it is an accelerated version), but it does not achieve a better dependence on approximation factor when compared with accelerated IBP and APDAGD. However, it does have a better dependence on the size of the empirical distributions and barycenter, n, than these two methods. This tradeoff is not explored in the paper. Not hints as to interesting theory from the analysis of FastIBP are given. -The explanation of the algorithm is short, and instead of giving heuristic ideas they instead just give the steps of the algorithm. Can anything be learned from this method or is it actually just taking some past work and applying it to this problem? Why is it able to achieve the better dependencies over other methods? - The authors don't compare with some mentioned methods, accelerated IBP and APDAGD, in the experiments. - It is unclear whether the assumption that all measures are supported on the same number of points necessary for these results - is the extension easy based on these results? Does the algorithmic complexity result extend?

Correctness: The claims and methodology appear to be correct.

Clarity: While the paper is generally well written and manages to fit in a lot, there are a few ways in which it could be made more clear. - I think this one way this could be addressed by restructuring the paper - when reading Section 2.3, it is unclear how these results will be used later on especially since Section 3 immediately follows it. I assume that these results are used in the complexity bounds given in Section 4, but only the bound in Lemma 2.1 is referenced there. The connections should be spelled out explicitly, and maybe moved together so that it flows better. - The proof in Section 3 is hard to follow. The matrix R appears to not be defined until (8), but it is referencing the matrix in (7). What does it mean for the "rows of the matrix to be in the set I"? How were these sets I and the j set chosen - did you just search until a counterexample was found? Propositions 3.1 and 3.3 are referenced at the end of the proof, but they should be referenced earlier as they define the notions used within the proof, as language such as "Indeed, columns 1 and 2 imply that rows 1, 4 and 5 are in the same set." -- here you are not explicitly stating it, but it is because you are deriving a contradiction to Proposition 3.2. Therefore, a lot of the language and notation is implied by what is written around it instead of explicitly and directly stating what is meant. Cleaning this up would greatly help with comprehension of this proof. It seems like maybe some things got lost in summarizing/shortening a longer paper. - Please make sure everything is defined within the paper. For example, where is "normalized objective value defined" (as used in the experiments)?

Relation to Prior Work: It is clearly discussed how this work relates to past works.

Reproducibility: Yes

Additional Feedback: Overall, this paper tackles an important problem and makes a step towards a better understanding of a simpler case of the the Wasserstein barycenter problem. While showing the problem to not be a MCF, the authors still give an algorithm with complexity at least somewhat comparable to state of the art. I mentioned the main downsides in the weaknesses section, but in summary, these are: 1) the writing needs to improve, as it feels like much was left out and unclear in the shortening of the paper, 2) the algorithm is not tested in regimes of varying n to see how the algorithm scales compared to other existing algorithms, 3) the algorithm was not compared with other algorithms with good complexity (accelerated IBP and APDAGD) and it is not said why they are not compared. However, even with these shortcomings, I think that this work is important and interesting enough to warrant acceptance into NeurIPS, as it both solves an open question and gives a new algorithmic approach to this problem, which seems to perform well in the limited experiments given. #### POST REBUTTAL #### After seeing the authors response, I am satisfied that they address my main concerns. That is, they give some experiments in the regimes of varying n in the supplementary material, they discuss the novelty of the FastIBP method, and they will improve the discussion of the paper. Since I have already given a high score, I will leave it as it is.

[Author Response · NeurIPS 2020]

**We would like to thank the reviewers for their efforts on evaluating our paper.** We appreciate that they pointed
out the importance of the problem, and carefully checked our analysis. All the minor comments have been corrected as
suggested, and we commit to releasing the source code around september. We address some common concerns below,
and focus next on specific answers to each reviewer.

**A. Empirical evaluations:** Our current working draft has significantly improved numerical results. We cannot put them
into this one-page response, but describe them succinctly. We have conducted experiments on Wasserstein barycenter
problem for large document datasets with word embedding and report the results. Second, we add comparisons with
other baselines (see **Point B** for details). Finally, we will follow the recommendations of R3 in terms of wording.

**B. Baseline approaches:** We thank R3 and R5 for pointing out that ADMM, ProxIBP, APDAGD and accelerated IBP
should be included in the experiment. ProxIBP is a combination of IBP and proximal point method. The same strategy
can be applied to FastIBP so it seems better to compare ProxIBP and ProxFastIBP. We will include those comparisons
in the revision. For ADMM, APDAGD and accelerated IBP, we tried to implement them but found the performance is
not ideal. This might be due to implementation issues. For example, APDAGD and accelerated IBP both require the
line search procedure which is sensitive to parameter tuning. We will report our findings in more details in the revision.

**C. Extension to more general settings:** We can certainly extend our results without assuming that all measures are
supported on the same number of points. We assumed it to avoid heavier notations. It is true that the ideas of this paper
can not be generalized to the free-support Wasserstein barycenter problem. Indeed, the computation of free-support
barycenters requires solving a multimarginal OT problem where the complexity bounds of algorithms become much
worse; please refer to [2] for details.

In the following, we provide answers to specific questions raised by each referee.

**Reviewer 1.** We thank R1 for your evaluation. Please see **Point A** for empirical evaluations.

**Reviewer 3.** We thank R3 for your comments. Please see **Point A** for empirical evaluations and **Point B** for baseline
approaches.

♦ *Intuition behind steps 1 and 3 in Algorithm 2.:* These steps are inspired by the momentum step in Nesterov's
accelerated gradient method for convex smooth optimization. While the convergence property of IBP is only built on
the convexity of the dual objective function [1], steps 1-3 further exploits the smoothness of $\varphi$ to achieve the fast rate:
$\varphi(\check{\lambda}^t, \check{\tau}^t) - \varphi(\lambda^\star, \tau^\star) = O(t^{-2})$. This is a crucial step, which leads to our improved complexity bound in Theorem 4.3.

**Reviewer 4.** We thank R4 for your review. Please see **Point A** for empirical evaluations.

♦ *The proof of Theorems 3.5 and 4.3 are in the supplement.:* We will add two detailed remarks behind Theorems 3.5
and 4.3 to further elaborate the proof ideas in the main context of the revised version.

**Reviewer 5.** We thank R5 for your evaluation. Please see **Point A** for empirical evaluations, **Point B** for baseline
approaches and **Point C** for the extension of our approach.

♦ *The writing needs to improve, as it feels like much was left out and unclear in the shortening of the paper.:* We agree.
We will reorganize the paper by spelling out the connection between Sections 2.3 and 4 explicitly. We will also revise
the confusing parts in Section 3 as you suggested to improve its readability.

♦ *The algorithm is not tested in regimes of varying $n$ to see how the algorithm scales compared to other existing*
*algorithms.:* We have tested in regimes of varying $n$ and presented the results in the supplement material; see Table 2.

♦ *why can* FASTIBP *achieve such bound?:* We will clarify in the updated version that FASTIBP is not a trivial
combination: (i) steps 1-3 and 8 are standard in optimization literature yet first introduced to accelerated OT algorithms;
(ii) steps 4-7 are specialized to the fixed-support barycenter problem. Furthermore, accelerated IBP and APDAGD are
based on primal-dual framework, which allows for directly optimizing $E_t$ and thus achieves better dependence on $1/\varepsilon$
than FASTIBP. In contrast, FASTIBP optimizes $E_t$ through the dual objective gap, which leads to better dependence on
$n$. Investigating the relationship between these two frameworks is interesting and we will try to provide discussions in
the revision.

## References

[1] S. Guminov, P. Dvurechensky, N. Tupitsa, and A. Gasnikov. Accelerated alternating minimization, accelerated
Sinkhorn's algorithm and accelerated iterative Bregman projections. *ArXiv Preprint: 1906.03622*, 2019.

[2] T. Lin, N. Ho, M. Cuturi, and M. I. Jordan. On the complexity of approximating multimarginal optimal transport.
*ArXiv Preprint: 1910.00152*, 2019.


[Meta-Review · NeurIPS 2020]

This paper studies approximation of the fixed support Wasserstein barycenter problem. The question is how to find the barycenter of a set of discrete probability distributions. While optimal transport can be cast as a min-cost flow, the authors solve an open problem in showing that fixed support Wasserstein barycenter is not. The paper also proposes a fast approximation method and a few experimental results showing a tradeoff between speed and accuracy. Reviewers felt the experimental evaluation could be significantly better but overall this is a great contribution, suitable for Neurips.